# Correcting 3D cloud effects in $X_{CO2}$ retrievals from OCO-2

Steffen Mauceri[1], Steven Massie[2], Sebastian Schmidt[2]

[1]Jet Propulsion Laboratory, California Institute of Technology, Pasadena, CA, USA
[2]Laboratory for Atmospheric and Space Physics, University of Colorado, Boulder, Colorado 80303, USA

*Correspondence to*: Steffen Mauceri (Steffen.Mauceri@jpl.nasa.gov)

**Abstract.** The Orbiting Carbon Observatory-2 makes space-based radiance measurements in the Oxygen A-band and the Weak and Strong carbon dioxide ($CO_2$) bands. Using a physics-based retrieval algorithm these measurements are inverted to column-averaged atmospheric $CO_2$ dry-air mole fractions ($X_{CO2}$). However, the retrieved $X_{CO2}$ are biased due to calibration issues and mismatches between the physics-based retrieval radiances and observed radiances. Using multiple linear regression, the biases are empirically mitigated. However, a recent analysis revealed remaining biases in the proximity of clouds caused by 3D cloud
radiative effects (Massie et al., 2021) in the processing version B10. Using an interpretable non-linear machine learning approach, we develop a bias correction model to address these 3D cloud biases. The model is able to reduce unphysical variability over land and sea by 20% and 40%, respectively. Additionally, the 3D cloud bias corrected $X_{CO2}$ show agreement with independent ground-based observations from the Total Carbon Column Observation Network (TCCON). Overall, we find that the published OCO-2 data record underestimates $X_{CO2}$ over land by -0.3 ppm in the tropics and northward of 45° N. The
approach can be expanded to a more general bias correction and is generalizable to other greenhouse gas experiments, such as GeoCarb, GOSAT-3 and CO2M.

## 1 Introduction

The Orbiting Carbon Observatory OCO-2 (Eldering et al., 2017; Crisp et al., 2004) makes space-based top-of-atmosphere radiance measurements in three spectral bands: Oxygen A band at 0.76 µm, the Weak $CO_2$-band at 1.61 µm, and the Strong $CO_2$ band at 2.06 µm. Using an optimal estimation retrieval (Rodgers, 2000) called ACOS (O'dell et al., 2018), these measurements are converted to column-averaged atmospheric $CO_2$ dry-air mole fractions ($X_{CO2}$). ACOS employs a physics-based forward model that takes into consideration viewing and solar geometry and various atmospheric and surface parameters.
Since OCO-2 generates on the order of 100,000 soundings per day, ACOS makes multiple approximations to speed up the retrieval algorithm. Most importantly, the retrieval makes the independent pixel approximation, where the radiance in a given sounding only depends on the properties (e.g. surface pressure, surface reflectance, aerosols, trace gas concentration) within

the field of view of this sounding. This approximation exploits that for most clear sky observations there is no significant horizontal exchange of photons.

35         Nearby clouds, however, can scatter a significant number of photons into the field of view of OCO-2 which enhances the observed radiance. This horizontal exchange of photons due to clouds, or 3D cloud effect, is not accounted for in the ACOS retrieval. Nevertheless, the forward model attempts to match the enhanced radiances which leads to errors in the converged state vector and most importantly, negative biases in retrieved $X_{CO2}$ (Massie et al., 2021; Massie et al., 2017; Merrelli et al., 2015; Emde et al., 2022; Kylling et al., 2022; Yu et al., 2021). Merrelli et al. (2015) applied the Spherical Harmonics Discrete

Ordinate Method (SHDOM) 3D radiative transfer code (Evans, 1998) to perturb OCO-2 type spectra, and calculated OCO-2 retrievals without and with the 3D radiance perturbations. Retrieved $X_{CO2}$ values were *lower* than clear sky retrievals by 0.3, 3, and 5-6 ppm for surfaces characterized by bare soil, vegetation, and snow-covered footprints, respectively. From an empirical perspective, Fig. 6 of Massie et al. (2021) demonstrates that retrieved $X_{CO2}$ over sea generally decreases when the distance between observations and clouds becomes less than 5 km.

45         Nearby clouds can also cause radiance dimming due to cloud shadows. Cloud brightening occurs on both sides of clouds since 40% of OCO-2 observations are within 4 km of clouds (Massie et al., 2021), and cloud brightening extends over a 5 to 10 km horizontal scale. A cloud shadow occurs only on one side of a cloud, with the shadow covering a limited angular portion of the side. Since the majority of OCO-2 observations are next to low-level clouds (think of an observation embedded in low-level Amazon cloud streets), the cloud shadows project only about one km or so from the low-level clouds. Using a year's data

volume, Massie et al. (in prep.) discuss detailed calculations, based on an analysis of OCO-2 O2 A-band continuum radiances, that yield an estimate of cloud shadowing frequency to be on the order of 4%, compared to 96% for the observations influenced by cloud brightening.

        To mitigate biases in retrieved "raw" $X_{CO2}$, a linear bias correction and threshold-based filtering is applied to the data, yielding "biased corrected" $X_{CO2}$. Bias correction and filtering are based on co-retrieved elements from the state vector that

are used to bring retrieved $X_{CO2}$ into agreement with multiple truth sources (Kiel et al., 2019). These truth sources include a "small areas analysis" which assumes that $X_{CO2}$ is constant over small distances (<100 km) within the same orbit, comparisons to ground-based observations from the Total Carbon Column Observation Network (TCCON) (Wunch et al., 2010), and comparisons to a multi model-mean of six models that assimilate in-situ data. Nevertheless, there are remaining underestimates in retrieved $X_{CO2}$, that have been linked to 3D cloud effects, in the proximity of clouds with an average of -0.4 and -2.2 ppm

for high quality and low quality data (Massie et al., 2021). To address these biases Massie et al. (2021) developed a linear bias correction and filtering approach using a set of features indicative of 3D cloud effects calculated from Moderate Resolution Imaging Spectroradiometer (MODIS) and OCO-2 files. However, biases in $X_{CO2}$ caused by nearby clouds are highly non-linear. Consequently, the present study has two goals. The first goal is to explore if a non-linear bias correction can reduce 3D cloud biases further than a linear approach. While the developed cloud features (H3D, HC, CSNoiseRatio, Cloud Distance,

discussed below) more directly capture 3D cloud effects, co-retrieved variables from the state vector might be more indicative

of the resulting $X_{CO2}$ biases. Thus, the second goal is to investigate if additional variables, co-retrieved with $X_{CO2}$, can be used to further reduce 3D cloud biases.

## 2 Data

We make use of OCO-2 (B10) (https://disc.gsfc.nasa.gov/datasets/OCO2_L2_Lite_FP_10r/, last access: 05/2022) data from
September 2014 to July 2019. These files contain bias corrected $X_{CO2}$ for soundings over sea in glint mode (in which sunlight is directly reflected by the Earth's surface towards OCO-2) and soundings over land with a nadir viewing geometry. We correct for remaining 3D cloud biases by utilizing a variety of parameters describing the retrieved atmospheric state vector, viewing and solar geometry, results from OCO-2 cloud screening pre-processors, location and time, and a quality flag (QF) for each sounding. The QF is determined by a series of hand tuned thresholds for various variables derived from state vector elements
that are indicative of retrieval biases in $X_{CO2}$. High-quality data has a QF=0 and low-quality data has QF=1. Similarly, the operational bias correction is performed with hand tuned linear fits to various state vector elements (Kiel et al., 2019).

In addition, we utilize ground-based observations by TCCON from all 27 stations that are in close proximity in time (24 h) and space (2.5° in latitude, 5° in longitude) to OCO-2 observations (https://tccondata.org, last access: 05/2022). The ground-based observations are used for validation only. However, they can only provide comparisons for a limited number of locations,
with relatively few ground-based sites in the Tropics and island locations.

Finally, we make use of four variables indicative of 3D cloud effects (Massie et al., 2021): H3D, HC, CSNoiseRatio, and Cloud Distance. H3D (Liang et al., 2009; Massie et al., 2017) describes the normalized standard deviation of the MODIS radiance field, and is calculated based on off-line MODIS radiance data files (Cronk, 2018). The radiance standard deviation is calculated in a circle with a radius of 10 km surrounding each OCO-2 data point. HC is calculated from differences in O2
A-band continuum radiances of an observation point and adjacent points in three rows (frames) of footprints. A frame has eight adjacent OCO-2 footprints, with each footprint on the order of 2 km in size. CSNoiseRatio is the ratio of the O2 A-band continuum radiance spatial standard deviation and noise level, calculated within a footprint (which has 20 "ColorSlice" sub-pixel elements). These three variables are indicative of 3D cloud effects since radiance gradients are present when clouds are next to observation footprints (radiance enhancements become larger as cloud distance decreases). Cloud Distance (Massie et
al., 2021) is the distance of the nearest cloud to each observation point, as determined from off-line radiance data files (Cronk, 2018), which contain 500 m MODIS radiances, geolocation and cloud mask data. Calculated 3D cloud features can be found for OCO-2 from September 2014 to July 2019 at https://doi.org/10.5281/zenodo.4008764.

# 3 Methods

## 3.1 Small Areas and TCCON as Truth metric

As a pre-processing step we match the 3D cloud variables, OCO-2 soundings, and TCCON by time and location. Afterwards, we remove soundings where no 3D cloud variables are available. To develop the bias correction model, we use the small areas analysis, which is based on the assumption that $CO_2$ is a well-mixed gas and assumed to be constant over spatial scales of less than ~100 km (though, there can be exceptions for strong $CO_2$ emitters such as mega cities). To exploit this constraint on $X_{CO2}$ we split OCO-2 soundings from the same orbit into small areas with a maximum size of 100 km. Each small area is generated by collecting soundings (sorted by observation time) until the distance between the first and last sounding exceeds the 100 km threshold. Afterwards, the collection process of the next small area is started. For each small area we identify soundings that are assumed to be free of 3D cloud biases (nearest cloud is at least 10 km away). From those soundings we define the median retrieved $X_{CO2}$ as the true $X_{CO2}$ of a given small area and any differences to this median are treated as biases. Small areas that contain less than 10 soundings free from 3D cloud biases are removed from the dataset. Since this process biases the remaining small areas towards longer cloud distances, we resample the remaining soundings so that the distribution of nearest cloud distances is similar to the original data set with about 40% of the sounding having a nearest cloud distance of less than 4 km. Note, this processing will interpret real $X_{CO2}$ enhancements, for example from power plants, as positive biases. However, we postulate that these cases are rare and that a model that is robust to outliers can still learn a useful bias correction from these data. Next, we remove outliers with large $X_{CO2}$ errors from the data set by applying a series of thresholds to the variables from the state vector. The variables and their thresholds are given in Table 1. Note that these filters remove only a small fraction of soundings (4%). Finally, we remove small areas with fewer than 20 soundings. This results in approximately $5 \cdot 10^6$ soundings over land and $20 \cdot 10^6$ soundings over the ocean, with a small subset of the soundings having coincident TCCON measurements. TCCON can only provide comparison for a limited set of regions with most stations in the Northern Hemisphere and over land. This challenges the development of a bias correction approach based on $X_{CO2}$ - TCCON differences that would be representative of areas far away from existing stations, such as Africa, South America and most of the ocean. Therefore, we use TCCON only as an independent truth metric for validation and not to develop the model itself.

**Table 1: Variables and their thresholds used to remove outliers**

| Variable | Description | Land | Sea |
|---|---|---|---|
| co2_ratio | Ratio of retrieved $X_{CO2}$ in $WCO_2$ and $SCO_2$ bands | x < 1 or x > 1.04 | x < 1 or x > 1.03 |
| co2_grad_del | Change between the retrieved $CO_2$ profile and the a priori profile | x < -100 or x > 100 | x < -50 or x > 100 |
| deltaT | Retrieved offset to a priori temperature profile | | x < 0 |
| dpfrac | Retrieved $X_{CO2}$ multiplied by difference in retrieved and a priori surface pressure (Kiel et al., 2019) | x > 7 | |
| rms_rel_sco2 | Root Mean Squared error of the L2 fit residuals for the $SCO_2$ band, relative to the continuum signal | | x > 0.5 |
| snr_sco2 | Signal-to-noise ratio in $SCO_2$ band | | x < 200 |

130    The distribution of nearest cloud distance, biases from the small area analysis and comparison to TCCON for land nadir and sea glint observations with QF=0 and QF=1 are shown in Figure 1. The plots show that the majority of OCO-2 soundings are taken within close proximity of clouds and that many of those soundings are filtered out in the current OCO-2 product (QF=1). This is especially problematic for areas such as the tropics that are dominated by clouds and, as a result, have few valid soundings. The small area and TCCON biases for QF=0 data are roughly normally distributed with a mean and standard

135    deviation of $0.0 \pm 0.5$ ppm for small area biases and $0.2 \pm 0.8$ ppm compared to TCCON for soundings over sea. For soundings over land the small area bias and bias compared to TCCON are similar with a mean and standard deviation of $0.1 \pm \sim 1$ ppm. For QF=1 the distribution of biases have a larger standard deviation for small area biases (land: 2.7 ppm, sea: 1.8 ppm) and compared to TCCON (land: 3.7 ppm, sea: 1.9 ppm), are skewed, and contain negative biases that far exceed positive biases, as analysed with the small areas (land: -0.6 ppm, sea: -1.2 ppm) and compared to TCCON (land: -1.2 ppm, sea: -1.2 ppm).

140    This long tail distribution of negative biases is indicative of 3D cloud effects (Massie et al., 2021) and should be mitigated with a successful 3D cloud bias correction.

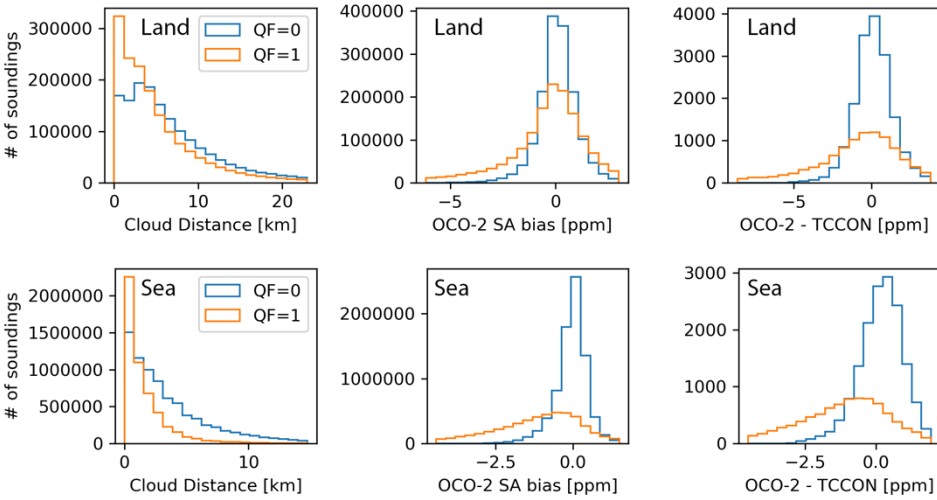

Figure 1: Histogram of data used in this study for nearest cloud distance (left), small area (SA) biases (middle), and biases compared to TCCON (right) for soundings over land (top) and sea (bottom). Higher quality data (QF=0) is shown in blue, lower quality data (QF=1) in orange.

## 3.2 Train-, Validation-, Test-split

To fit, or *train*, the bias correction model we used soundings from September 2014 to the end of July 2017, totalling roughly $12 \cdot 10^6$ and $3 \cdot 10^6$ soundings over sea and land, respectively. To find the best model parameters and evaluate what features minimize biases the furthest we use a separate validation set containing soundings from the beginning of August 2017 to the end of July 2018. Finally, to test how the trained model performs on new data we use a separate testing set of soundings from the beginning of August 2018 to the end of July 2019. The validation and testing set have $3 \cdot 10^6$ and $2 \cdot 10^6$ soundings over the sea with QF=0 and QF=1, respectively, and $5 \cdot 10^5$ soundings each over land with QF=0 and QF=1.

## 3.3 Bias Correction Model

We train two types of models for the bias correction, non-linear models (Random Forest) and linear models (Ridge Regression) to provide a baseline comparison. A Random Forest is an ensemble of classifying decision trees and outputs the mean of those trees (Breiman, 2001). Random Forests are easy to interpret and robust to outliers. Each tree is trained in a supervised manner with a random subset (50%) of the available training data, also referred to as bootstrapping. Using the training data, each tree iteratively splits the data using the feature that can minimize the mean squared error of the predictions the furthest, until it reaches a maximum user-provided number of splits, or *depth*. For our land model we used a depth of 8 and for our ocean model a depth of 15. The larger model size for the ocean is mostly due to there being more training data available over the ocean than

over land which allows to fit a larger model that still generalizes to new data. Each random forest was composed of 100 individual trees. These parameters were chosen to maximize model performance on the validation set. The model inputs are a set of selected features from the OCO-2 retrieved state vector (e.g. co2_grad_del) and the model output is the remaining $X_{CO2}$ bias derived from the small areas analysis.

Since the operational OCO-2 bias correction uses a linear approach, we also perform a baseline comparison to a linear model. We choose multi-variate linear regression with a small Tikhonov regularization term (the regularization helps if some of the inputs are correlated, which is the case for most real-world applications), also referred to as ridge regression (Hoerl and Kennard, 1970a, b). Thus, using the training set we seek to find the weights, $w$, that minimize the following equation:

$$\|y - Xw\|_2^2 + \alpha\|w\|_2^2 \tag{1}$$

where $y$ is the standardized (mean removed and divided by standard deviation) $X_{CO2}$ bias, $X$ are the standardized features, $\|\cdot\|_2$ is the Euclidean norm, and $\alpha$ controls the strength of the Tikhonov regularization. For our application we found $\alpha = 10^{-5}$ to maximize performance on the validation set.

### 3.4 Feature Selection

First, we identified retrieved state variables that show a strong dependence (change in mean or variability) to nearest cloud distance, indicating that they might be good candidates to correct for 3D cloud effects. Two examples are shown in Figure 2. In addition to the list of identified features we added solar and viewing geometries and surface albedo. Those variables have a direct physical impact on 3D cloud effects; 3D cloud effects are amplified at large solar zenith angles and for brighter surfaces (Okata et al., 2017). Finally, we removed highly correlated variables. This results in a set of 23 features for soundings over land, and 24 features for soundings over sea, that may be used to correct for 3D cloud biases in retrieved $X_{CO2}$ (more information about each variable can be found on pages 29 to 40 in (Jet Propulsion Laboratory, 2018)). Next, we used *recursive feature elimination* to identify what subset of features can reduce biases the furthest. Reducing the number of features makes the model more robust to new data, avoids *overfitting,* and aids interpretability.

For the recursive feature elimination, we removed one feature at a time, trained a small random forest model with 32 trees each on a random selection of $5 \cdot 10^5$ soundings with QF=0 and QF=1 from the training set. Afterwards we calculated the model performance on the full validation set. As the performance metrics we used the correlation coefficient ($R^2$) between modelled bias and existing bias as indicated by the small-areas calculations. The feature, that has been removed from the highest performing model, is then permanently removed and the process is repeated until only one feature is left. The iterative process was performed separately for land and sea soundings. The order of the feature elimination and resulting $R^2$ is shown in Figure

3. The least important variables are shown at the top and were removed first. A low importance can either result from a variable varying independently of biases in $X_{CO_2}$ or the variable could be correlated with another variable (e.g. dp and dp_abp) or set of variables that provide similar information, making one of them obsolete. The most important variables are shown on the bottom.

For our bias-correction model we decided to use the five most important variables for land and four most important variables for sea soundings, as identified by the feature elimination. These variables explain most of the variance and partially overlap for land and sea. For land the most important variables are **dp_abp** (retrieved surface pressure from pre-processor retrieval, minus surface pressure from the GEOS-5 FP-IT model), **h2o_ratio** (ratio of retrieved $H_2O$ column from the $WCO_2$ band to that from the $SCO_2$ band), **co2_grad_del** (a measure of the difference in the retrieved and prior $CO_2$ vertical gradient), **dp** (retrieved surface pressure from the L2 Full-Physics retrieval, minus the O2A-band prior surface pressure), and **aod_water** (retrieved extinction optical depth of cloud water at 755 nm). For sea the most important variables are dp, co2_grad_del, **aod_ice** (retrieved extinction optical depth of cloud ice at 755 nm), and **albedo_wco2** (retrieved Lambertian albedo in the $WCO_2$ band). Note that the final set of features does not include any of the 3D cloud metrics used in the bias correction discussed in Massie et al. (2021). Additionally, solar and viewing geometry were removed in the iterative process. However, the process includes the surface albedo in the weak $CO_2$ band, dp, and dp_abp which have a direct physical connection to 3D cloud effects. As discussed below in relation to Fig. 2b, dp_abp and nearest cloud distance are empirically correlated. Additionally, increased values in aod_water and deviations from unity for h2o_ratio are indicative of cloud contamination (Jet Propulsion Laboratory, 2018). This indicates that elements of the operational retrieval state vector (co2_grad_del, dp, dp_abp, h2o_ratio, aod_water, aod_ice, albedo_wco2) are more directly correlated with remaining biases in $X_{CO_2}$ (due to 3D cloud and other effects) than features that directly measure 3D cloud effects which perturb the radiation field (H3D, HC, CSNoiseRatio).

From an operational standpoint, using elements from the current retrieval state vector to correct 3D cloud biases simplifies the bias correction in future operational products. It also is more generally applicable to other missions that might not have available coincident cloud field measurements, that can be applied to derive nearest cloud distances, such as OCO-3 (Eldering et al., 2019). On the other hand, it reduces the interpretability of the developed model and does not allow to directly link 3D cloud biases to 3D cloud metrics. The OCO-2 and 3D cloud variables and their meaning are summarized in Table 2.

Note that it is not possible to clearly separate biases due to 3D cloud effects and other mismatches between the forward model of the retrieval algorithm and the observed radiances. For example, differences in modelled and real aerosol optical properties (Chen et al., 2022) or uncertainties in absorption profiles of various trace gases (Payne et al., 2020) likely are important. Additionally, uncertainties in the instrument calibration can cause systematic biases as well. Thus, some of the features might also correct for non-3D cloud effects. However, we tried to mitigate the effect of non-3D cloud biases by only adding features to the feature selection process that show some dependence to nearest cloud distance (see Figure 2) or have a direct physical relationship to 3D cloud biases. Additionally, our bias correction is applied to data that has already been corrected with the operational OCO-2 bias correction (our processing utilizes bias corrected $X_{CO_2}$). Thus, biases independent to 3D cloud effects should be minimized.

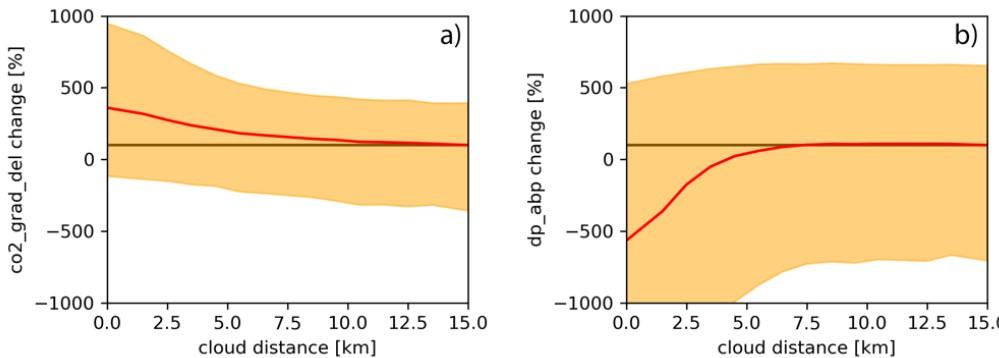

Figure 2: Change of variability and mean in percent of potential features with respect to nearest cloud distance. Change in mean is shown in red; change of the 5th and 95th percentile is shown in yellow; no change (baseline) is shown with a brown straight line. Change is calculated with respect to feature mean for observations with a nearest cloud distance of 14 km to 15 km. a) co2_grad_del, and b) dp_abp. Please refer to the text for a description of the two features.

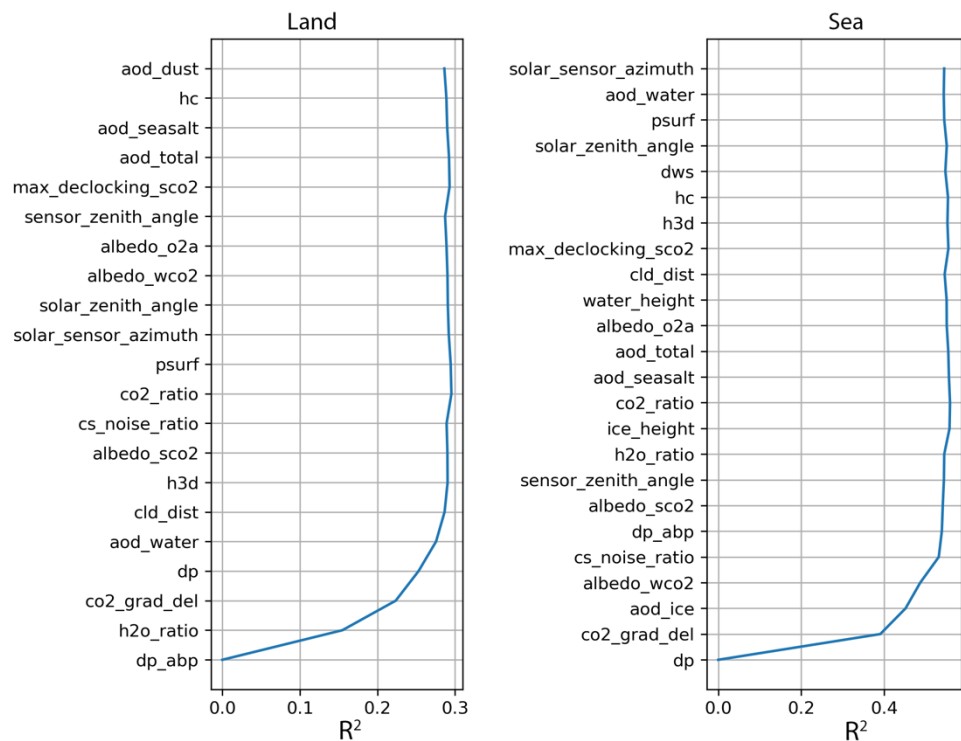

Figure 3: Feature ordering by importance as determined by recursive feature elimination. Features were removed from top to bottom with the most important features on the bottom. The model performance for removing a given feature is indicated with $R^2$

240

calculated on the validation set. Please refer to the pages 29 to 40 in (Jet Propulsion Laboratory, 2018) for a description of the individual features.

**Table 2: Summary of OCO-2 state vector variables and 3D cloud variables**

| Variables | Description |
|---|---|
| dp_abp | Retrieved surface pressure from pre-processor retrieval, minus surface pressure from forecast model |
| dp | Retrieved surface pressure from full-physics retrieval, minus surface pressure from $O_2A$-band prior |
| h2o_ratio | Ratio of retrieved $H_2O$ column from $WCO_2$ band to $SCO_2$ band |
| co2_grad_del | Change between retrieved $CO_2$ profile and a priori profile |
| aod_water | Retrieved extinction optical depth of cloud water |
| aod_ice | Retrieved extinction optical depth of cloud ice |
| albedo_wco2 | Retrieved surface albedo in $WCO_2$ band |
| | |
| H3D | Normalized standard deviation of the radiance field |
| HC | Differences in continuum radiances of an observation to adjacent observations |
| CSNoiseRatio | Ratio of continuum radiance spatial standard deviation and noise level |
| Cloud Distance | Distance to the nearest cloud |

## 4 Results

### 4.1 Reduction in $X_{CO2}$ biases

After the random forest was trained using the training set (09/2014 – 07/2017) we evaluated the model performance on the testing set (08/2018 – 07/2019). Figure 4 compares remaining $X_{CO2}$ biases in OCO-2 (as determined by the small areas analysis) with biases after our correction is applied (OCO-2 corr.) for QF=0 and QF=1 soundings. For land soundings $X_{CO2}$ biases are reduced from a Root Mean Square Error (RMSE) of 2.0 ppm to 1.6 ppm (see Figure 4c). For sea soundings the bias correction has a significantly bigger impact and reduces biases from 1.4 ppm to 0.9 ppm (see Figure 4d). Over the sea the bias correction mostly corrects negative biases less than -0.8 ppm (see Figure 4b).

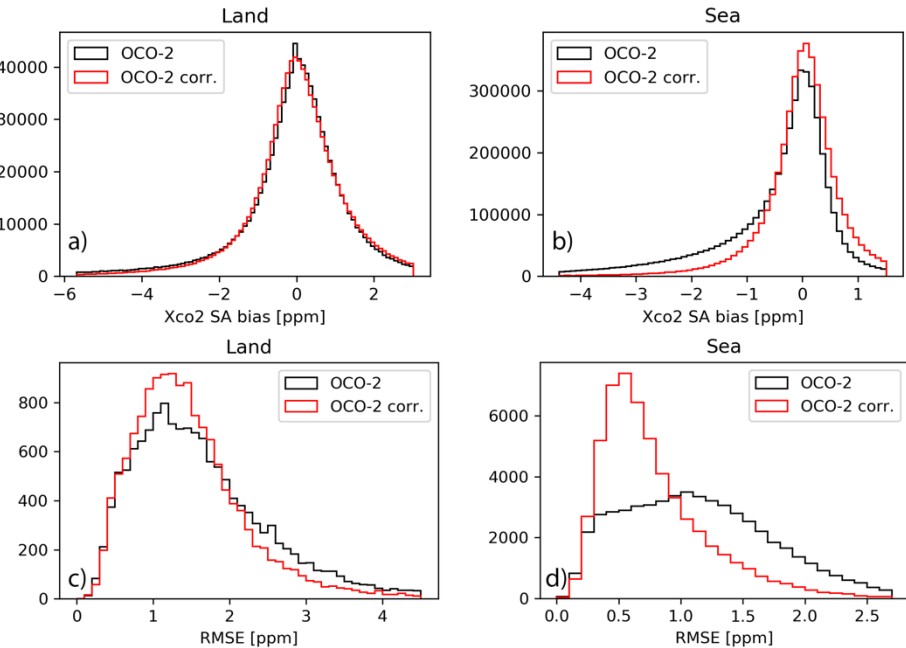

**Figure 4: Reduction in non-physical variability in $X_{CO2}$ for OCO-2 and the proposed bias correction approach (OCO-2 corr.) for land (left) and sea (right) for QF=0 and QF=1 data from 2018 to 2019. (Top) distribution of biases from individual soundings; (bottom) distribution of standard deviation for individual small areas.**

Table 3 shows the RMSE by quality flag. For QF=0 and QF=1 the biases in $X_{CO2}$ corrected with our model are less than the testing set operational OCO-2 biases. However, for QF=0 improvements by our correction (OCO-2 corr.) compared to OCO-2 are small (<10%). The QF=0 data have significantly fewer soundings with clouds in close proximity (see Figure 1) which explains in part the smaller difference. Additionally, the quality flags are determined so that the operational linear bias correction of OCO-2 works well i.e., $X_{CO2}$ biases have a mostly linear relationship to elements of the state vector where QF=0. For QF=1 the difference is more significant, reducing the RMSE from 2.6 ppm to 2.1 ppm over land and from 2.3 ppm to 1.3 ppm over sea.

**Table 3: RMSE of $X_{CO2}$ as determined by small areas analysis for the testing set (08/2018 – 07/2019). The RMSE is shown for the operational OCO-2 product (OCO-2), the proposed bias correction approach (OCO-2 corr.), a linear bias correction using the same**

features than the proposed approach (OCO-2 lin. corr.), and a random forest using dedicated cloud metrics (OCO-2 cloud corr.). The data is separated by high quality data (QF=0), low quality data (QF=1), and all data (QF=0 + 1).

| | Land $X_{CO2}$ [ppm] | | | | Sea $X_{CO2}$ [ppm] | | | |
|---|---|---|---|---|---|---|---|---|
| | OCO-2 | OCO-2 corr. | OCO-2 lin. corr. | OCO-2 cloud corr. | OCO-2 | OCO-2 corr. | OCO-2 lin. corr. | OCO-2 cloud corr. |
| QF=0 | 0.94 | **0.90** | 0.92 | 0.92 | 0.53 | **0.44** | 0.49 | 0.46 |
| QF=1 | 2.60 | **2.10** | 2.31 | 2.43 | 2.29 | **1.31** | 1.52 | 1.44 |
| QF=0 + 1 | 1.95 | **1.62** | 1.79 | 1.88 | 1.42 | **0.89** | 1.06 | 0.98 |

To more directly link the bias correction to 3D cloud effects we show biases with respect to nearest cloud distance in Figure 5. $X_{CO2}$ from OCO-2 shows a clear negative mean bias and increased variance for a nearest cloud distance of less than 3 km and 4 km over land and sea, respectively. After applying our bias correction the mean bias in the proximity of clouds is close to zero. Thus, the bias correction effectively mitigates biases due to 3D cloud effects.

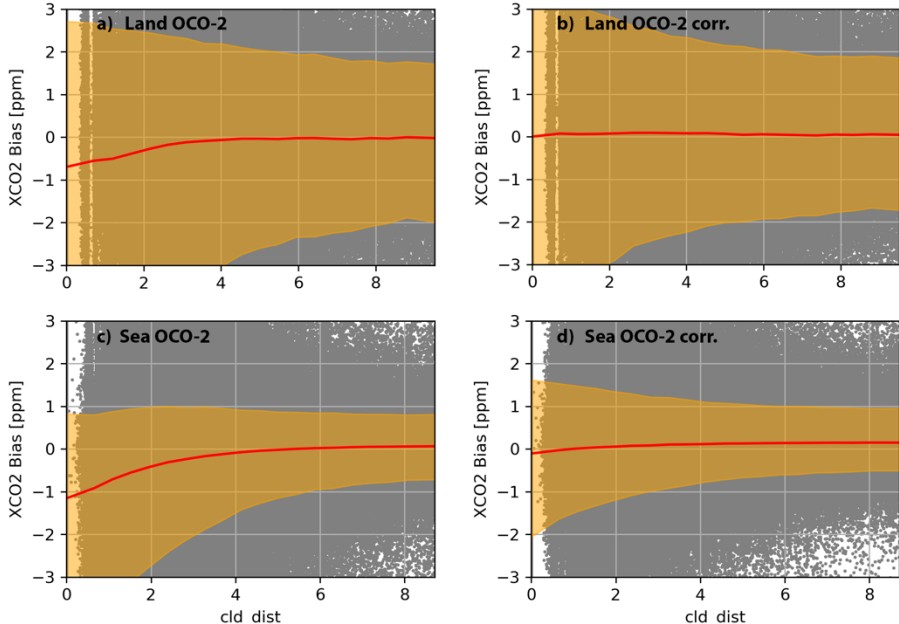

**Figure 5: $X_{CO2}$ bias vs cloud distance for OCO-2 soundings over land (a), soundings over land corrected by our proposed method (b), OCO-2 soundings over sea (c), and soundings over sea that are corrected (d) for QF=0 and QF=1 data from 2018 to 2019. The 5th and 95th percentiles are indicated with the yellow shaded area; the mean is shown with a red line and individual comparisons with grey dots.**

## 4.2 Linear vs Non-linear bias correction

Building on the work by Massie et al. (2021) one of the guiding research questions was whether a non-linear approach based on interpretable machine learning techniques would improve upon a linear 3D cloud bias correction. To probe this question, we compare the performance of the non-linear random forest model to linear ridge regression (see Equation 1). To train the linear model we used the same features, training and testing sets used for the random forest model development. The RMSE for the linear model (OCO-2 lin. corr.) and non-linear model (OCO-2 corr.) are shown in Table 3. For QF=0 land and sea observations the linear and non-linear model have similar performance with the non-linear model allowing for a slightly lower RMSE. For QF=1 the non-linear random forest reduces remaining biases further than the linear ridge regression from 2.3 ppm to 2.1 ppm over land and from 1.5 ppm to 1.3 ppm over the sea.

## 4.3 Comparison to Using Dedicated Cloud Variables

A second question that we wanted to answer was whether additional variables from the OCO-2 retrieved state vector could improve the 3D cloud bias correction. As shown in Figure 3, the four cloud variables (H3D, HC, CSNoiseRatio, Nearest Cloud Distance) were removed during the recursive feature elimination step, indicating that other variables form the state vector are more directly correlated with $X_{CO2}$ biases. To better understand how much of the model performance stems from the new set of features we performed a set of experiments. For the first experiment we trained a random forest using only the four cloud variables in addition to surface albedo, solar zenith angle, sensor zenith angle, and the difference between solar and sensor azimuth. The results are shown in Table 3 (OCO-2 cloud corr.). As expected, using the cloud variables with the non-linear random forest model performs worse than using the random forest with the features identified using the recursive feature elimination. One caveat of this experiment is that our bias correction approach, aimed at 3D cloud biases, might also make corrections for biases stemming from other effects (e.g. aerosols) that are independent to clouds and, thus, cannot be explained with cloud variables. Unfortunately, clearly separating various sources of bias is not possible.

For the other experiment we combine the 3D cloud variables with the variables determined by the recursive feature elimination (dp_abp, co2_grad_del, h2o_ratio, dp, and aod_water for land and dp, co2_grad_del, aod_ice, and albedo_wco2 for sea) and compare the results to using only the features from the recursive feature elimination. If adding the 3D cloud variables would significantly reduce biases in $X_{CO2}$ further it would indicate that the set of identified features is mostly correcting for biases unrelated to 3D cloud effects. In total we compare the model performance of four sets of features: the features determined by the recursive feature elimination in addition to a) nearest cloud distance, b) CSNoiseRatio, c) nearest cloud distance, CSNoiseRatio, HC, H3D, and d) deltaT (see Table 1 and 2). The last set of features serves as a control experiment where we quantify the effect of adding a random variable that is unrelated to 3D cloud effects to the set of chosen features. The results are shown in Table 4. For QF=0 there are practically no differences for the four test cases compared to our chosen set of features. For sea QF=1 data the best set of features is c) which reduces the RMSE from 1.31 ppm to 1.25 ppm. Overall, the addition of 3D cloud variables (a, b, c) allows the models to lower the RMSE further compared to our

proposed model, however, the improvements are only marginal. This indicates that the set of chosen features in our bias correction model accounts for the majority of 3D cloud biases in $X_{CO2}$. Further evidence for this is shown in Figure 5 and is presented in the next section with an independent comparison to TCCON.

**Table 4: RMSE of $X_{CO2}$ as determined by small areas analysis for the testing set (08/2018 – 07/2019). The RMSE is shown for the**
**proposed bias correction approach (OCO-2 corr.) and using the same approach but with additional features. In addition to the variables determined by the recursive feature elimination a) contains nearest cloud distance, b) CSNoiseRatio, c) nearest cloud distance, CSNoiseRatio, HC, H3D, and d) deltaT.**

| | Land $X_{CO2}$ [ppm] | | | | | Sea $X_{CO2}$ [ppm] | | | | |
|---|---|---|---|---|---|---|---|---|---|---|
| | OCO-2 corr. | a) | b) | c) | d) | OCO-2 corr. | a) | b) | c) | d) |
| QF=0 | **0.90** | 0.90 | 0.90 | 0.90 | 0.90 | **0.44** | 0.43 | 0.43 | 0.42 | 0.44 |
| QF=1 | **2.10** | 2.09 | 2.10 | 2.08 | 2.10 | **1.31** | 1.29 | 1.28 | 1.25 | 1.28 |
| QF=0 + 1 | **1.62** | 1.62 | 1.62 | 1.61 | 1.62 | **0.89** | 0.87 | 0.86 | 0.84 | 0.87 |

**4.4 Comparison to TCCON**

We further compare bias corrected $X_{CO2}$ to TCCON. TCCON observations have low uncertainties and are used to validate OCO-2 retrieved $X_{CO2}$. However, they can only provide point measurements and are non-uniformly distributed, with most TCCON sites over land and in the Northern Hemisphere. For our comparison we consider coinciding observations of OCO-2 and TCCON for the period of the testing set (08/2018 - 07/2019). This results in 1768 (QF=0: 7459, QF=1: 2794) matches
over land and 1305 (QF=0: 2165, QF=1: 1111) matches over sea. Note that our bias correction model was trained without taking TCCON observations into consideration while OCO-2 takes OCO-2 – TCCON biases explicitly into consideration for its linear bias correction, filtering, and to calculate global offsets. Thus, comparisons between OCO-2 and TCCON are not independent.

Table 5 shows the mean and standard deviation of differences between OCO-2 and TCCON and after we apply our bias
correction (OCO-2 corr. - TCCON) for QF=0 and QF=1. Over land and sea the bias correction reduces the standard deviation between OCO-2 and TCCON for QF=0 and QF=1. For observations over sea the bias corrected $X_{CO2}$ exhibits a systematic positive offset compared to TCCON of about 0.7 ppm. The systematic offset could be addressed by recalculating the scaling factor used for retrievals over sea in OCO-2. However, there are only few TCCON stations that can provide comparisons for those data and these stations are not equally distributed over the ocean.


**Table 5: Mean and standard deviation of bias in $X_{CO2}$ compared to TCCON observations for the testing set (08/2018 – 07/2019). The comparison for the operational OCO-2 product is indicated by (OCO-2 - TCCON) and the proposed random forest approach by (OCO-2 corr. - TCCON).**

|  | Land $X_{CO2}$ [ppm] | | Sea $X_{CO2}$ [ppm] | |
|---|---|---|---|---|
|  | OCO-2 - TCCON | OCO-2 corr. - TCCON | OCO-2 - TCCON | OCO-2 corr. - TCCON |
| QF=0 | -0.05 ± 1.24 | **-0.16 ± 1.21** | 0.57 ± 0.74 | **0.67 ± 0.69** |
| QF=1 | -1.7 ± 4.21 | **0.45 ± 2.77** | -1.01 ± 1.83 | **0.59 ± 1.34** |
| QF=0 + 1 | -0.47 ± 2.42 | **-0.23 ± 1.74** | 0.06 ± 1.40 | **0.69 ± 0.97** |


To better understand how the bias correction addresses 3D cloud biases as compared to TCCON, Figure 6 shows $X_{CO2}$ biases vs nearest cloud distance. For land and sea there exist negative biases in OCO-2 in the proximity of clouds (Figure 6a and 6c). Interestingly, there is a positive bias for OCO-2 sea data when no clouds are close to OCO-2 soundings (> 2 km) that likely stems from OCO-2 incorporating a multi model mean in its bias correction in addition to TCCON. After applying our bias

correction, $X_{CO2}$ biases over land show a reduced dependence on nearest cloud distance(Figure 6b). For sea, the bias correction pushed $X_{CO2}$ up by roughly 0.5 ppm in the proximity of clouds, resulting in a uniform positive bias of 0.7 ppm independent of cloud distance (Figure 6d). Thus, the bias correction removed the dependency of $X_{CO2}$ biases on nearest cloud distance but did not address the overall offset present in OCO-2.

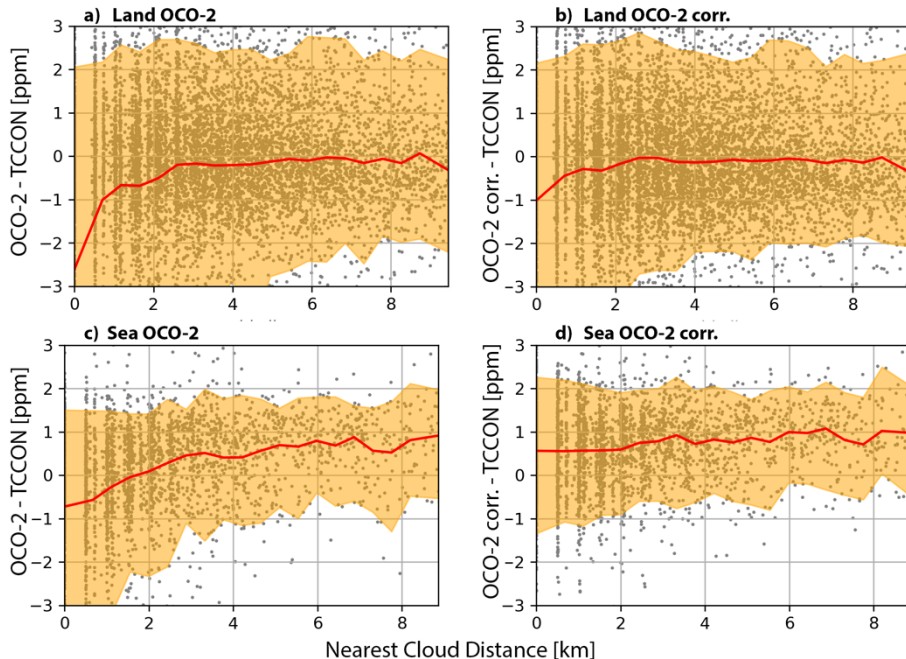

 **Figure 6: X$_{CO2}$ bias vs cloud distance of OCO-2 over land (a), OCO-2 corrected (b), OCO-2 over sea (c), and OCO-2 corrected (d), for QF=0 and QF=1 data from 2018 to 2019. The 5$^{th}$ and 95$^{th}$ percentiles are indicated with the yellow shaded area, the mean is shown with a red line, and individual comparisons with grey dots.**

## 5 Discussion

### 5.1 Model Interpretation

To better understand how the model utilizes the input features to calculate the bias correction we show the modelled biases with respect to the individual features in Figure 7. Overall, the bias-feature relationship is non-linear for most features over the complete state space but linear over part of the state space. This explains the lower model performance of the linear model we compared to in Section 4.2. over the complete state space (QF=0+1) and the only marginal improvement compared to QF=0 data. Differences between retrieved surface pressure and a reference surface pressure (**dp**, **dp_abp)** show a positive correlation

with X$_{CO2}$ biases. When the operationally retrieved surface pressure is underestimated, X$_{CO2}$ is underestimated as well. The ratio of the retrieved H$_2$O column from the WCO$_2$ band to that from the SCO$_2$ band (**h2o_ratio),** for soundings over land (Figure 7b), is independent to X$_{CO2}$ biases for ratios of less than one and has a strong negative correlation for ratios above one. A ratio of 1.05 corresponds on average to an X$_{CO2}$ bias of -1.5 ppm. The difference between the retrieved CO$_2$ profile and the a priori profile (**co2_grad_del**) shows mostly a positive correlation for negative values (surface CO$_2$ is underestimated

compared to CO$_2$ higher up in the atmosphere) and a negative correlation for positive values. This indicates that 3D cloud effects challenge the accurate retrieval of the X$_{CO2}$ profile. The sensitivity of X$_{CO2}$ biases to changes in co2_grad_del is approximately twice as strong over sea than land (see Figure 7c and 7f). This feature cannot be exclusively linked to 3D cloud effects since it is one of the most important features for the operational bias correction of OCO-2. The retrieved extinction optical depth of cloud water (**aod_water)** shows a mostly negative linear correlation with a X$_{CO2}$ bias of -2 ppm for an

extinction optical depth of 0.1. The retrieved extinction optical depth of cloud ice (**aod_ice)** is negatively correlated with X$_{CO2}$ biases. Finally, the surface albedo in the weak CO$_2$ band (**albedo_wco2)** has mostly no dependence to X$_{CO2}$ biases for most of its range but shows some negative correlation with biases for brighter surfaces. Note that our bias correction is applied in addition to the bias correction that has already been performed in the operational OCO-2 retrieval. While the operational OCO-2 bias correction does not explicitly account for 3D cloud biases it might implicitly mitigate such biases with its linear bias

correction (since the operational bias correction variable dP is correlated to nearest cloud distance, see the red line in Fig. 2b).

To understand why some variables of the OCO-2 retrieved state vector are correlated with 3D cloud biases it is important to remember that the operational retrieval, based on optimal estimation, tries to match the observed radiances with a forward radiative transfer model. However, while the observed radiances can be perturbed by 3D cloud effects, the forward model tries to match those radiances with an independent pixel approximation that does not physically include 3D cloud effects. In

particular the 3D cloud effect enhances, or brightens, the radiances as compared to no clouds being present. To compensate for this brightening the forward model decreases the retrieved surface pressure (reduction in dp and dp_abp), increases the optical depth of cloud water (aod_water) and increases the surface albedo in the WCO$_2$ band. These relationships are shown

empirically in Figure 7. As shown in Fig. 2 of Massie et al. (2021), the spectral signature of the 3D cloud effect (the optical depth structure of the radiative perturbation of the 3D effect) differs from the spectral signatures of perturbations in surface pressure, surface reflectivity, aerosol, and $X_{CO2}$. Fig. 2 illustrates that a decrease in surface pressure and $X_{CO2}$, and an increase in surface reflectance will increase the observed radiance. In order to provide for extra radiance enhancement in the cloud brightened observed radiance, a variety of state variable adjustments (and their unique spectral contributions) are utilized by the retrieval to bring forward model radiances in agreement with the observed radiances. The relationship of 3D cloud biases to surface pressure differences and surface albedo are likely due to a combination of physically-based 3D cloud radiative effects and operational retrieval algorithmic considerations.

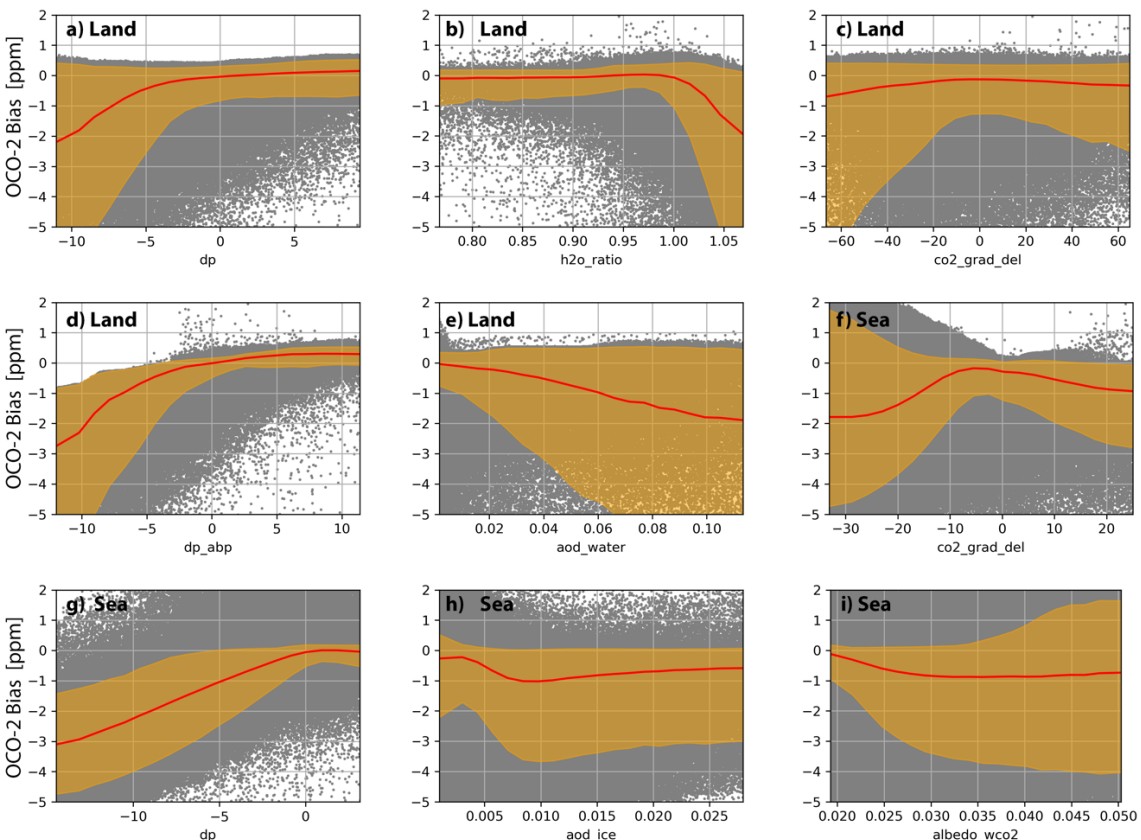

**Figure 7: Bias identified by correction by the proposed model (OCO-2 Bias) with respect to its features: dp (a), h2o_ratio (b), co2_grad_del (c), dp_abp (d), aod_water (e) over land and co2_grad_del (f), dp (g), aod_ice (h), and albedo_wco2 (i) over sea for QF=0 and QF=1 data from 2018 to 2019. The 5th and 95th percentile are indicated with the yellow shaded area, the mean is shown with a red line and individual comparisons with grey dots. The scale of the x-axis for each plot is different. Please refer to Section 3.4 for a description of the individual features.**

A further look at the relative importance of the model features shows dp_abp being the most important feature for land and dp for sea observations. Over land dp_abp is followed by dp, h2o_ratio, co2_grad_del, and aod_water. Over sea dp is followed by co2_grad_del, aod_ice, and albedo_wco2 (see Figure 8). The feature importance was calculated as the normalized total reduction of mean square error brought by an individual feature. I.e., if we were to omit dp from our model as a feature the bias correction would be less effective than if we were to omit co2_grad_del.


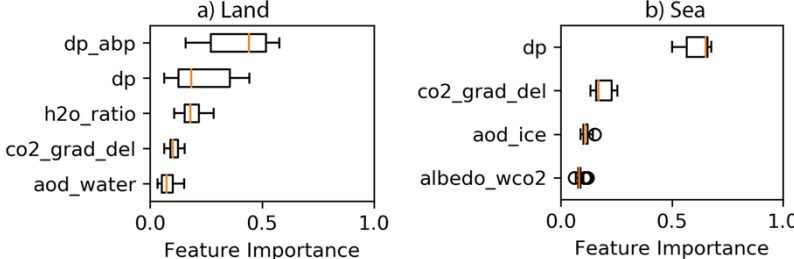

**Figure 8: Feature importance for the bias correction model. Feature importance is shown for land (left) and sea (right) observations. Model was trained using the training set with QF=0 and QF=1 data. Please refer to Section 3.4 for a description of the individual features.**


## 5.2 Regional Biases

To further understand regional impacts of our bias correction we calculate biases, as identified by our model, for soundings from 2014 to 2019 and averaged results over 2° by 2° cells (see Figure 9). I.e., in applying the proposed bias correction, the results shown in Figure 9 are subtracted from OCO-2 $X_{CO2}$. Since using soundings only from the testing set leads to many

areas with no data, we used all available data (2014 - 2019) for this visualization. Over land negative biases (i.e., $X_{CO2}$ from OCO-2 is underestimated) are present north of 45° in America, Europe and Asia, averaging -0.3 ppm. Around the tropics within ±10° of the equator, average biases are near -0.3 ppm as well. Positive biases are most dominant over the deserts of northern Africa and Saudi Arabia. Over sea biases are more equally distributed than over land. When comparing the regional biases to a map of nearest cloud distance (see Figure 10) there is overlap between negative biases and areas dominated by

clouds (correlation coefficient between nearest cloud distance and OCO-2 bias is R=0.3).

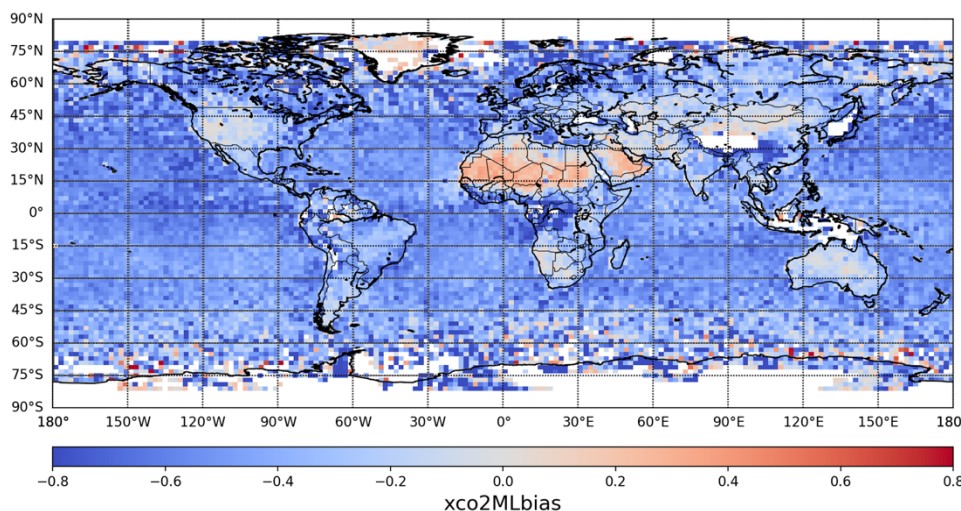

**Figure 9: Biases in $X_{CO_2}$ identified by our model. Biases are averaged over 2° by 2° for all soundings (2014 to 2019, QF=0 and QF=1). Negative biases are shown in blue, positive biases in red, no data in white.**


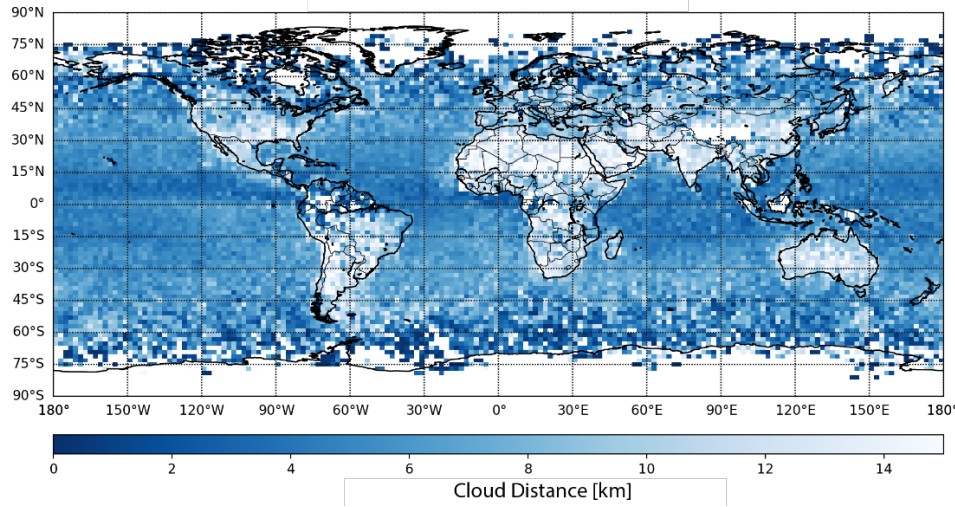

**Figure 10: Nearest cloud distance derived from MODIS. Nearest cloud distances are averaged over 2° by 2° for all matched soundings (2014 to 2019, QF=0 and QF=1). Darker blues indicate closer clouds, no data is shown in white.**


### 5.3 Effect of bias correction on true CO₂ Enhancements

As discussed in section 3.1 we use the small areas analysis as a truth proxy to develop our model. This assumes that $CO_2$ is well mixed and constant over short spatial scales (<100 km). However, this assumption is violated for strong $CO_2$ emitters

such as power plants. Even though these strong emitters are rare in the data and likely don't influence the bias correction

model, there is a risk that the model would "correct", i.e. remove real local $CO_2$ enhancements. To confirm that real $CO_2$ enhancements are still present after the proposed bias correction, we compare OCO-2 retrieved and corrected $X_{CO2}$ from three OCO-2 overpasses over large coal power plants (see Figure 11), that have been used in a previous study (Nassar et al., 2017). The $CO_2$ enhancements of the retrieved and corrected $X_{CO2}$ for the three overpasses (the singular spikes in $X_{CO2}$ in the middle of the graphs) agree closely and demonstrates that the *bias correction does not erroneously remove true $CO_2$ enhancements*

*from the OCO-2 data record*.

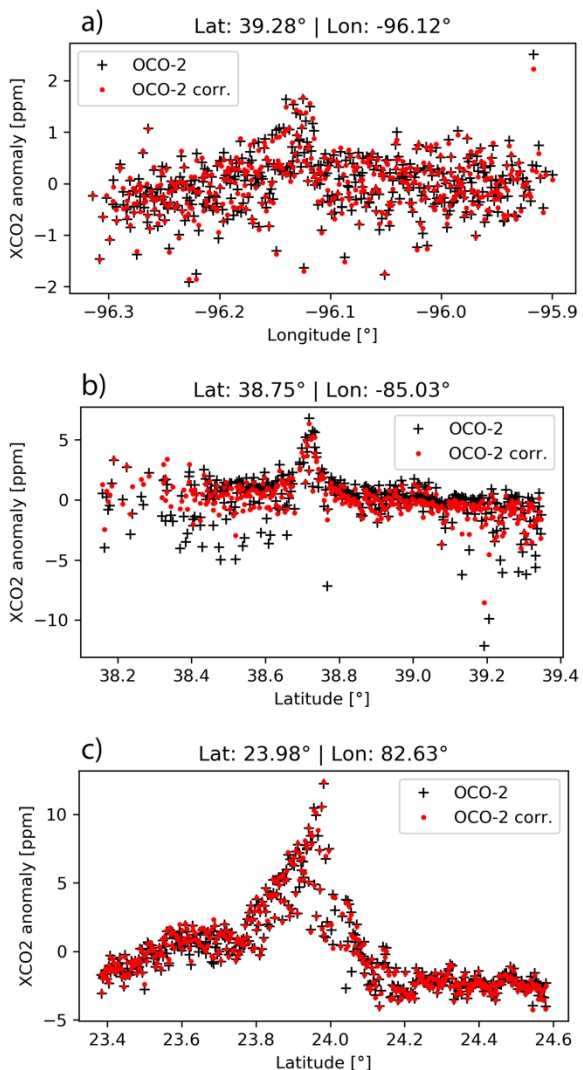

**Figure 11: $X_{CO2}$ anomalies for OCO-2 and bias corrected OCO-2 retrievals in the proximity of coal power plants. Power plant a) Westar at Lat: 39.28° Lon: -96.12° on 12/04/2015, b) Ghent at Lat: 38.75° Lon: -85.03° on 08/13/2015, c) Sasan at Lat: 23.98° Lon: -**

**82.63° on 10/23/2014. Anomaly is calculated by subtracting the mean.**

## 5 Future Work and Conclusion

### 5.1 Future Work

The developed bias correction approach is aimed at mitigating 3D cloud biases in OCO-2, but could readily be expanded to a more general bias correction. Future research will need to show how far the approach used in this research (determining the bias correction solely from small area biases) will work for correcting previously uncorrected $X_{CO2}$ (the "raw" XCO2 from the operational retrieval). For such a correction a two-step approach might be necessary that combines a global (comparison to TCCON) and local (small area analysis) bias correction approach. However, developing such an approach would be challenged by the sparse coverage of TCCON stations.

The operational bias correction used for OCO-2 is aimed at QF=0 data. This is highlighted by the significant reduction in $X_{CO2}$ biases our correction was able to achieve on QF=1 data while improvements in QF=0 data where small. Filtering out low quality data is a simple approach to improve the overall quality of the OCO-2 $X_{CO2}$ retrieval. However, it leaves certain areas with too few samples, most notably the tropics (due to clouds), higher latitudes (due to large solar zenith angles) and around Brazil, Bolivia, Paraguay (due to the South Atlantic Anomaly). Improving the bias correction of future OCO-2 versions that allow for less restrictive filtering would benefit applications that rely on those data.

Finally, one could expand the approach taken here, developing one model for land and one for sea data, to having multiple models for land and sea to better capture the diverse causes for biases in $X_{CO2}$ across Earth, for example, different types of aerosols dominate different areas and might lead to specific biases in different regions or seasons. Such a location based bias correction could also be expanded to a location-based filtering approach that would, for example, allow less restrictive filtering at higher latitudes (Mendonca et al., 2021; Jacobs et al., 2020) to have more of those soundings pass the filter and be available for scientific inquiry. A key challenge of such an approach will be validation due to the limited number of available TCCON stations.

### 5.2 Conclusion

We identified five variables from the state vector for OCO-2 retrievals over land (dp, dp_abp, h2o_ratio, co2_grad_del, aod_water) and four variables over sea (dp, co2_grad_del, aod_ice, albedo_wco2) that are used in a machine learning model that allows to mitigate 3D cloud biases in OCO-2 retrieved $X_{CO2}$. We demonstrate that this machine learning model does not erroneously remove true $CO_2$ power plant enhancements from the OCO-2 data record. All variables are bi-products of the operational retrieval used by OCO-2 which simplifies their inclusion for bias correction in future versions of the operational product. The proposed non-linear bias correction is based on a random forest approach and able to reduce the RMSE from 1.95 ppm to 1.62 ppm over land and 1.42 ppm to 0.89 ppm over sea for QF=0 and QF=1 data on an independent testing set. We demonstrated a systematic approach to correct for biases in optimal estimation retrievals. Namely, (1) find a physical

variable that is well understood and correlated with the cause of the bias (in our case 'nearest cloud distance'). (2) Identify elements from the retrieved state vector and other retrieval products that show a dependence to the variable from step (1) in addition to other variables that have a physical connection to the bias. (3) Use recursive feature elimination to identify which

subset of the elements identified in (2) should be used for the bias correction. (4) Use a simple explainable machine learning model to map the features identified in (3) to the biases and correct for them.

**Author contribution**

SMAU, SMAS, and SS conceptualized the research goals. SMAU and SMAS prepared the various datasets. SMAU developed the approach, implemented the experiments and visualized the results. SMAU prepared the manuscript with contributions from all co-authors.

The authors declare that they have no conflict of interest.

**Acknowledgement**

We acknowledge support by NASA grant 80NSSC21K1063 "Mitigation of 3D cloud radiative effects in OCO-2 and OCO-3 $X_{CO2}$ retrievals". We appreciate the TCCON teams, who measure and provide ground-based $X_{CO2}$ validation to the carbon cycle research community (Dubey et al., 2014a; Dubey et al., 2014b; Iraci et al., 2014; Iraci et al., 2016; Toon and Wunch, 2014; Wennberg et al., 2016a; Wennberg et al., 2014c; Wennberg et al., 2014a; Wennberg et al., 2016b; Wennberg et al.,

2014b; Wunch et al., 2017; Wunch et al., 2015).

The research was carried out at the Jet Propulsion Laboratory, California Institute of Technology, under a contract with the National Aeronautics and Space Administration (80NM0018D0004).

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
