# Peer review of "Correcting 3D cloud effects in $X_{CO2}$ retrievals from OCO-2"

_Atmospheric Measurement Techniques, 2022_

## Author Comment (AC1)

We thank the anonymous reviewer for taking the time to review this manuscript and providing helpful feedback. Your advice has been very helpful and lead to significant changes to the manuscript that improved the overall quality.

The supplement contains the reviewer comments in black and our responses are shown in red *Italic* with citations from the manuscript in " ".

Reviewer 1:

The authors develop a bias correction algorithm for the OCO-2 $XCO_2$ product using a (non-linear) random forest model with the aim to correct 3D cloud effects. The model is trained with bias derived from a "small area analysis" assuming no variation of $XCO_2$ at local scale (< 100 km). Variables from the OCO-2 CO2 dataset and dedicated cloud features are considered as model features. A feature selection is conducted to identify 3 and 4 features for glint and nadir, respectively, that are used for training the final model. The dedicated cloud features are excluded at this step. The analysis of the model shows a reduction of biases in the validation dataset and compared to TCCON measurements.

The manuscript is well written and the topic is within the scope of AMT. A correction of 3D cloud effects in $XCO_2$ retrievals would be an important scientific progress, because it would increase the number of observations with good quality. Thus, it would make $CO_2$ satellites better suited for studying anthropogenic and natural sources. However, I am skeptical that the presented machine-learning model will correct only non-physical variability due to cloud effects. Based on the current description of the method, I suspect that the model also wrongly corrects true variability in $XCO_2$, for example, from anthropogenic and natural sources (megacities, power plants, wildfires, etc.). My major concerns are the following:

**Data filtering**: The model is currently trained with QF=0 and QF=1 retrievals. The latter also includes retrievals where the quality is poor, for reasons other than (3D) cloud effects (e.g., high SZA, AOD, snow cover), which affects training, analysis and validation, because correlation coefficient and standard deviation are sensitive to outliers. I therefore think it is necessary to still apply some filtering to the QF=1 retrievals to remove poor-quality retrievals due to non-cloud effects.

*We fully agree and added a step in our data preparation pipeline where we remove outliers from the dataset as described below:*

*L108 "Next, we remove outliers with large $X_{CO2}$ errors by applying a series of thresholds to the variables from the state vector. The variables and their thresholds are given in Table 1. Note, that these filters remove only a small fraction of soundings (4%) and are not comparable to the quality flags used by OCO-2.*

*Table 1: Variables and their thresholds used to remove outliers*

| Variable | Description | Land | Sea |
|---|---|---|---|
| co2_ratio | Ratio of retrieved $X_{CO2}$ in WCO2 and SCO2 bands | *x < 1 or x > 1.04* | *x < 1 or x > 1.03* |
| co2_grad_del | Change between the retrieved $CO_2$ profile and the a priori profile | *x < -100 or x > 100* | *x < -50 or x > 100* |

| | | |
|---|---|---|
| *deltaT* | *Retrieved offset to a priori temperature profile* | *x < 0* |
| *dpfrac* | *Retrieved XCO2 multiplied by difference in retrieved and a priori surface pressure (Kiel et al., 2019)* | *x > 7* |
| *rms_rel_sco2* | *Root Mean Squared error of the L2 fit residuals for the SCO2 band, relative to the continuum signal* | *x > 0.5* |
| *snr_sco2* | *Signal-to-noise ratio in SCO2 band* | *x < 200* |

*"*

**Truth metric**: The generation of the truth metric is described insufficiently. Apparently, a k-mean algorithm is used to divide OCO-2 orbits into small areas where XCO2 does not vary strongly. The true bias is then defined as the deviation of the XCO2 retrieval from the median in this area. I would like to see some examples for some orbits to better judge how well this method works. In particular, it is unclear what happens in proximity of CO2 sources (megacities and power plants) where XCO2 deviates from the mean due to local emissions. I assume that a filtering algorithm needs to be applied to remove these areas to avoid false (positive) biases in the training dataset, but such filtering is not mentioned in the manuscript.

**Model validation**: The validation needs to be able show that true $XCO_2$ enhancements are not wrongly corrected by the model. Since TCCON is not well suited for this task as the instrument are generally not located downstream of a source, I suggest conducting some case studies near known $CO_2$ sources to show the effect of the bias correction in OCO-2 data. There is a large amount of literature on the use of OCO-2 to estimate power plant emissions with suitable cases (e.g., Nassar et al. 2017, Hakkarainen et al 2021). The validation should also be done for the B10-cloud model.

*We simplified the generation of the small areas to enhance transparency of how they are generated:*

*L100 "To exploit this constraint on $X_{CO2}$ we split OCO-2 soundings from the same orbit into small areas with a maximum size of 100 km. Each small area is generated by collecting soundings (ordered by observations time) until the distance between the first and last sounding exceeds the 100 km threshold. Afterwards, the collection process of the next small area is started."*

*Power plant sources are indeed interpreted as positive biases in XCO2. However, they are rare and we assume that they don't affect the model significantly. We added this disclaimer to the description of the small areas and added an analysis of three overpasses over power plants to the manuscript to back up our claim:*

*L106 "Additionally, this processing will interpret real $X_{CO2}$ enhancements, for example from power plants, as positive biases. However, we postulate that these cases are rare and that a model that is robust to outliers can still learn a useful bias correction from these data."*

*L439 "**5.3 Effect of bias correction on true $CO_2$ Enhancements***

*As discussed in section 3.1 we use the small areas analysis as a truth proxy to develop our model. This assumes that $CO_2$ is well mixed and constant over short spatial scales (<100 km). However, this assumption is violated for strong $CO_2$ emitters such as power plants. Even though these strong emitters*

*are rare in the data and likely don't influence the bias correction model, there is a risk that the model would "correct", i.e. remove real local $CO_2$ enhancements. To confirm that real $CO_2$ enhancements are still present after the proposed bias correction, we compare OCO-2 retrieved and corrected $X_{CO2}$ from three OCO-2 overpasses over large coal power plants (see Figure 11), that have been used in a previous study (Nassar et al., 2017). The $CO_2$ enhancements of the retrieved and corrected $X_{CO2}$ for the three overpasses (the singular spikes in $X_{CO2}$ in the middle of the graphs) agree closely and demonstrates that the bias correction does not erroneously remove true $CO_2$ enhancements from the OCO-2 data record.*

[Figure]

***Figure 11: XCO2 anomalies for OCO-2 and bias corrected OCO-2 retrievals in the proximity of coal power plants. Power plant a) Westar at Lat: 39.28° Lon: -96.12° on 12/04/2015, b) Ghent at Lat: 38.75° Lon: -85.03° on 08/13/2015, c) Sasan at Lat: 23.98° Lon: -82.63° on 10/23/2014. Anomaly is calculated by subtracting the average. "***

**Feature selection**: The feature selection method ignores that some variables could correlate with the "truth metric", which is computed from the same dataset and might have some issues (see previous point). These variables cannot be used in the model. In particular, the presented model uses "xco2_strong_idp" as

feature, which is $XCO_2$ retrieved from the IMAP-DOAS in pre-processing "normalized by subtracting the mean of each small area" (L155). This is extremely similar to the $XCO_2$ bias used for training, i.e. the difference between the $XCO_2$ retrieved from ACOS and the mean of each small area (L77). As a result, I strongly suspect that the bias correction correct not only cloud biases, but also any deviation from the local mean including enhancements due local sources (e.g. megacities and power plants). Applying this bias correction model to the OCO-2 $CO_2$product, would make it impossible to estimate accurate $CO_2$ emissions from OCO-2 observations. To avoid this issue, the features used in the model need to be selected based on their correlation with cloud properties. The B10-cloud model shown in Section 4.3 is likely a good choice. It could be given more emphasis in the manuscript.

*We agree that XCO2_strong_idp should be correlated with true changes in $X_{CO2}$ and removed the variables as a feature for our bias correction. Thanks for catching that. However, the other selected variables from the state vector should not be correlated with true variations in XCO2 over spatial scales of less than 100 km. Therefore, we did not remove those features. A bias correction based only on cloud properties has been the topic of a previous study by Massie et al. (2021), thus, we don't go into too much detail in this manuscript.*

**Specific comments**

L17: Not clear if you find the bias in the $XCO_2$ product with or without (cloud) bias correction.

*Clarified sentence. It now reads:*

*L18 "Overall, we find that the published OCO-2 data record underestimates $X_{CO2}$ … "*

L33: The effect of 3D cloud effects on TROPOMI $NO_2$ was recently studied: Emde et al. 2022, Yu et al. 2021, Kylling et al. 2022.

*L38 Added citation for the three papers*

L62ff: It would be nice to provide some more details on the 3D cloud effect features, so the reader does not need to check the cited citations.

*Added additional information:*

*L82 "Finally, we make use of four variables indicative of 3D cloud effects (Massie et al . 2021): H3D, HC, CSNoiseRatio, and Cloud Distance. H3D (Liang et al., 2009; Massie et al., 2017) describes the normalized standard deviation of the MODIS radiance field, and is calculated based on the Cronk (2022) off-line MODIS radiance data files. The radiance standard deviation is calculated in a circle of radius 10 km surrounding each OCO-2 data point. HC is calculated from differences in O2 A-band continuum radiances of an observation point and adjacent points in three rows (frames) of footprints. A frame has eight adjacent OCO-2 footprints, with each footprint on the order of 2 km in size. CSNoiseRatio is the ratio of the O2 A-band continuum radiance spatial standard deviation and noise level, calculated within a footprint (which has 20 "colorslice, CS" sub-pixel elements). These three variables are indicative of 3D cloud effects since radiance gradients are present when clouds are next to observation footprints (radiance enhancements become larger as cloud distance decreases). Cloud Distance (Massie et al., 2021) is the distance of the nearest cloud to each observation point, as determined from the Cronk (2022) off-line radiance data files,*

*which contain 500 m MODIS radiances, geolocation and cloud mask data. Calculated 3D cloud features can be found for OCO-2 from September 2014 to July 2019 at https://doi.org/10.5281/zenodo.4008764.”*

L67: Since overpass time would matter for "CloudDistance", please specify if you use MODIS Aqua and/or Terra.

*The MODIS Aqua data used to determine CloudDistance is acquired six minutes after the OCO-2 measurements.*

L95: Please provide reasoning why and how 3D cloud effects cause negative biases.

*Line 386 “In particular the 3D cloud effect enhances, or brightens, the radiances as compared to no clouds being present. To compensate for this brightening the forward model decreases the retrieved surface pressure (reduction in dp_abp), increases the optical depth of cloud water (aod_water) and increases the surface albedo in the WCO$_2$ band. These relationships are shown empirically in Figure 7 As shown in Fig. 2 of Massie et al. (2021), the spectral signature of the 3D cloud effect (the optical depth structure of the radiative perturbation of the 3D effect) differs from the spectral signatures of perturbations in surface pressure, surface reflectivity, aerosol, and $X_{CO2}$. Fig. 2 illustrates that a decrease in surface pressure and $X_{CO2}$, and an increase in surface reflectance will increase the observed radiance. In order to provide for extra radiance enhancement in the cloud brightened observed radiance, a variety of state variable adjustments (and their unique spectral contributions) are utilized by the retrieval to bring forward model radiances in agreement with the observed radiances. The relationship of 3D cloud biases to surface pressure differences and surface albedo are likely due to a combination of physically-based 3D cloud radiative effects and operational retrieval algorithmic considerations. “*

L101: Since OCO-2 might drift in time, please check if splitting by time affects your conclusions.

*A possible drift of OCO-2 or nature (climate change, XCO2 increasing steadily, ...) is exactly why a train test split by time is so important. Such a drift would negatively affect the accuracy and validity of our model. Thus, by using soundings from one time period to train the model and then a later time period to test the model we get an honest assessment how the model generalizes to new data (including any instrument drift).*

L143: Since the correlation coefficient is not sensitive to a bias in your model, it would be useful to use also other parameters (e.g. RMSE).

*Interesting idea. We repeated the recursive feature elimination with RMSE and got the same ordering of the most important features. This is not surprising since the random forest algorithm internally uses the mean square error to construct the individual trees.*

L180ff: The results here depend strongly on the definition of the truth metric, which might not contain only "non-physical variability" (see general point).

*That is a valid caveat of our analysis. We already have a disclaimer in In Section 3.1 where we note:*

*L98 “To develop the bias correction model, we use the ‘small areas analysis’, which is based on the assumption that CO$_2$ is a well-mixed gas and assumed to be constant over spatial scales of less than ~100 km **(there can be exceptions for strong CO$_2$ emitters such as mega cities).”***
*Added disclaimer to 4.1 L245: “Figure 4 compares remaining $X_{CO2}$ biases in OCO-2 **(as determined by the small areas analysis)** with biases after our correction is applied (OCO-2 corr.)”.*

L299ff: The analysis here assumes that the truth metric is caused primarily caused by 3D cloud effect, which is likely a wrong assumption (see general points). I think that it is necessary that you describe quite clearly here why a feature would be effect by 3D cloud effects, e.g., why ACOS retrieves a wrong surface pressure (dp) in the proximity of clouds.

*It is correct that separating biases caused by 3D cloud effects and other retrieval biases (e.g. due to aerosols) is challenging. Our plots in Figure 5 indicate that at a minimum we remove systematic negative biases in XCO2 in the proximity of clouds which indicates that we correct for 3D cloud effects. More theoretical work has been done by Massie et al. (2021) to show that these biases stem indeed from 3D cloud effects and not, for example, cloud shadows or other effects. Based on this work we added some information of how the individual variables relate to 3D cloud biases.*

*Line 383 "To understand why some variables of the OCO-2 retrieved state vector are correlated with 3D cloud biases it is important to remember that the operational retrieval, based on optimal estimation, tries to match the observed radiances with a forward radiative transfer model. However, while the observed radiances can be perturbed by 3D cloud effects, the forward model tries to match those radiances with an independent pixel approximation that does not physically include 3D cloud effects. In particular the 3D cloud effect enhances, or brightens, the radiances as compared to no clouds being present. To compensate for this brightening the forward model decreases the retrieved surface pressure (reduction in dp_abp), increases the optical depth of cloud water (aod_water) and increases the surface albedo in the $WCO_2$ band. These relationships are shown empirically in Figure 7. As shown in Fig. 2 of Massie et al. (2021), the spectral signature of the 3D cloud effect (the optical depth structure of the radiative perturbation of the 3D effect) differs from the spectral signatures of perturbations in surface pressure, surface reflectivity, aerosol, and $X_{CO2}$. Fig. 2 illustrates that a decrease in surface pressure and $X_{CO2}$, and an increase in surface reflectance will increase the observed radiance. In order to provide for extra radiance enhancement in the cloud brightened observed radiance, a variety of state variable adjustments (and their unique spectral contributions) are utilized by the retrieval to bring forward model radiances in agreement with the observed radiances. The relationship of 3D cloud biases to surface pressure differences and surface albedo are likely due to a combination of physically-based 3D cloud radiative effects and operational retrieval algorithmic considerations. "*

**Technical corrections**

L11 (and others): CO2 -> $CO_2$

*Corrected*

L26: fraction -> fractions

*Corrected*

L33 (and others): The citations style does not follow AMT requirements (here: "Massie et al. 2017")

*Updated citation style throughout the manuscript*

L118: form -> from

*Corrected*

**References**

- Emde et al. (2022) https://doi.org/10.5194/amt-15-1587-2022
- Hakkarainen et al. (2021) https://doi.org/10.1016/j.aeaoa.2021.100110
- Kylling et al. (2022) https://doi.org/10.5194/amt-15-3481-2022
- Nassar et al. (2017) https://doi.org/10.1002/2017GL074702
- Yu et al. (2021) https://doi.org/10.5194/amt-2021-338

---

## Author Comment (AC2)

We thank the anonymous reviewer for taking the time to review this manuscript and providing helpful feedback. Your advice has been very helpful and lead to significant changes to the manuscript that improved the overall quality.

The supplement contains the reviewer comments in black and our responses are shown in red *Italic* with citations from the manuscript in " ".

Reviewer 2:
**General comments**

This manuscript is about algorithmic correction for biases in the CO2 mixing ratio (XCO2) retrieved from OCO-2 spectra in data processor version B10. These biases are said to be caused by 3D cloud effects.

The manuscript is very technical. It uses many "jargon" terms which are only understandable in the OCO-2 algorithm and data processing context. The paper is difficult to read for non-experts.

*We simplified the language throughout the manuscript to make it more accessible to non-experts. Here are a few examples:*

- *Removed mentioning of the current processing version B10 for most of the manuscript and simply refer to OCO-2*
- *Removed mentioning of the quality flags (QF=0 and QF=1) where appropriate*
- *Added explanation for what ocean glint and land nadir observations are and simply refer to them as observations over sea and land throughout the manuscript*
- *Removed most mentions of the OCO-2 specific 'lite files'*
- *Simplified the algorithm and description of how the small areas are generated.*

The physics of 3D cloud effects is missing.

*Clarified description of 3D cloud effect:*

*L35 "Nearby clouds, however, can scatter a significant number of photons into the field of view of OCO-2 which enhances the observed radiance. This horizontal exchange of photons due to clouds, or 3D cloud effect, is not accounted for in the ACOS retrieval. Nevertheless, the forward model attempts to match the enhanced radiances which leads to errors in the converged state vector and most importantly, negative biases in retrieved $X_{CO2}$ (Massie et al., 2021; Massie et al., 2017; Merrelli et al., 2015; Emde et al., 2022; Kylling et al., 2022; Yu et al., 2021). Merrelli et al. (2015) applied the Spherical Harmonics Discrete Ordinate Method (SHDOM) 3D radiative transfer code (Evans 1998) to perturb OCO-2 type spectra, and calculated OCO-2 retrievals without and with the 3D radiance perturbations. Retrieved XCO2 values were lower than clear sky retrievals by 0.3, 3, and 5-6 ppm for surfaces characterized by bare soil, vegetation, and snow-covered footprints. From an empirical perspective, Fig. 6 of Massie et al. (2021) demonstrates that ocean $X_{CO2}$ generally decreases in value when the distance between observations and clouds becomes less than 5 km.*
*    Nearby clouds can also cause radiance dimming due to cloud shadows. But only 25 % of observed radiances pass into the operational retrieval, since two cloud pre-processors (Taylor et al. 2016) exclude many observed radiances. Cloud brightening occurs on both sides of clouds since 40% of OCO-2 observations are within 4 km of clouds (Massie et al. 2021), and cloud brightening extends over a 5 to 10*

*km horizontal scale. A cloud shadow occurs only on one side of a cloud, with the shadow covering a limited angular portion of the side. Since the majority of OCO-2 observations are next to low-level clouds (think of an observation embedded in low-level Amazon cloud streets), the cloud shadows project only about a km or so from the low- level clouds. Using a year's data volume, Massie et al. (in prep.) discuss detailed calculations, based on an analysis of OCO-2 O2 A-band continuum radiances, that yield an estimate of cloud shadowing frequency to be on the order of 4%, compared to 96% for the observations influenced by cloud brightening."*

The wider usefulness of this manuscript for other satellite instruments than OCO-2 is debatable.

*The findings of this manuscript are important for a series of current and upcoming missions. Most importantly, the manuscript shows that we can correct for 3D cloud biases without relying on coincident observations of cloud fields. This is important for missions that are not in the A-train, such as OCO-3. Furthermore, the manuscript shows that a non-linear bias correction can outperform a linear approach if the biases are non-linear themselves. This approach has more general applications for future missions such as GeoCarb, GOSAT-3 or CO2M.*

A major question - not answered by the paper - is whether the biases in XCO2 that are presented are due to real 3D cloud effects or due to other OCO-2 algorithmic deficiencies, other retrieval model deficiencies, or even due to L1 calibration effects. The title of the paper is thus a conjecture. The paper does not contain retrieval simulations of 3D cloud effects on XCO2 retrieval that would legitimate the title.

*We explain in the paper that the OCO-2 algorithm does not account for the 3D cloud effect in its retrieval. Thus, one would expect that the retrieval needs to somehow account for the brightening by the 3D cloud effect without taking the effect itself into consideration. This is where the biases stem from. We further show that the biases increase with decreasing distances to clouds (see Fig. 2b of the paper) Retrieval simulations are beyond the scope of this paper and will be presented in Massie et al. (in prep.)*

*Retrieval simulations have already been performed by Merrelli et al. (2015). We added some more information about this work:*

*Line 39 "Merrelli et al. (2015) applied the Spherical Harmonics Discrete Ordinate Method (SHDOM) 3D radiative transfer code (Evans 1998) to perturb OCO-2 type spectra, and calculated OCO-2 retrievals without and with the 3D radiance perturbations. Retrieved XCO2 values were lower than clear sky retrievals by 0.3, 3, and 5-6 ppm for surfaces characterized by bare soil, vegetation, and snow-covered footprints. From an empirical perspective, Fig. 6 of Massie et al. (2021) demonstrates that ocean $X_{CO2}$ generally decreases in value when the distance between observations and clouds becomes less than 5 km."*

There is no mention of cloud shadows, whereas these are also 3D cloud effects.

*That is an excellent point. The importance of cloud shadows has been explored in work by Massie et al., (in prep.). They concluded that they don't contribute significantly to the OCO-2 observed radiances. We added the following information to the manuscript:*

*Line 45 "Nearby clouds can also cause radiance dimming due to cloud shadows. But only 25 % of observed radiances pass into the operational retrieval, since two cloud pre-processors (Taylor et al. 2016) exclude many observed radiances. Cloud brightening occurs on both sides of clouds since 40% of OCO-2 observations are within 4 km of clouds (Massie et al. 2021), and cloud brightening extends over a*

*5 to 10 km horizontal scale. A cloud shadow occurs only on one side of a cloud, with the shadow covering a limited angular portion of the side. Since the majority of OCO-2 observations are next to low-level clouds (think of an observation embedded in low-level Amazon cloud streets), the cloud shadows project only about a km or so from the low- level clouds.  Using a year's data volume, Massie et al. (in prep.) discuss detailed calculations, based on an analysis of OCO-2 O2 A-band continuum radiances, that yield an estimate of cloud shadowing frequency to be on the order of 4%, compared to 96% for the observations influenced by cloud brightening."*

There is no referencing to other literature than OCO-2 papers and reports. One would expect references to cloud correction in earlier satellite data, e.g. SCIAMACHY or GOSAT.

*We were not able to find literature discussing retrievals that are biased on 3D cloud effects beyond the cited literature about OCO-2. Please provide concrete examples and we would be happy to include those.*

In fact, the paper is now an OCO-2 technical report. Unless the paper is drastically revised, it is not acceptable for AMT.

*The paper describes an explorative study of how to mitigate 3D cloud effects. There is currently no plan to apply this algorithm operationally to OCO-2. While the developed model can't be directly applied to other missions, the general approach can. Furthermore, the paper shows that it is possible to correct for 3D cloud biases without requiring cloud field observations, which is important for current and future missions where such observations are not available. Taking our two reviewers comments into consideration we made significant changes to the manuscript and believe that it is a valuable contribution to AMT and the science community.*

*The original paper was extensively rewritten to be more readable, with less jargon and more explanations.*

**Recommendation**

Please improve the paper significantly. Add simulations that show that the biases are due to 3D effects of clouds. Clarify the physics behind the features used in the OCO-2 fitting algorithm, and their meaning and relevance for this bias correction work. Then you could make a new submission to AMT, which is a suitable journal for this type of research.

*Simulations of the 3D cloud effect are beyond the scope of this paper. Theoretical calculations and 3D radiative transfer simulations are addressed in another paper that will be submitted shortly by Massie et al. Based on this work we added some information of how the individual variables relate to 3D cloud biases. There are also existing simulations by Merrelli et al. (2015) that we had previously cited. We added some additional information about this and previous work to the manuscript:*

*L39 "Merrelli et al. (2015) applied the Spherical Harmonics Discrete Ordinate Method (SHDOM) 3D radiative transfer code (Evans 1998) to perturb OCO-2 type spectra, and calculated OCO-2 retrievals without and with the 3D radiance perturbations. Retrieved XCO2 values were lower than clear sky retrievals by 0.3, 3, and 5-6 ppm for surfaces characterized by bare soil, vegetation, and snow-covered footprints. From an empirical perspective, Fig. 6 of Massie et al. (2021) demonstrates that ocean $X_{CO2}$ generally decreases in value when the distance between observations and clouds becomes less than 5 km."*

*L383 "To understand why some variables of the OCO-2 retrieved state vector are correlated with 3D cloud biases it is important to remember that the operational retrieval, based on optimal estimation, tries to match the observed radiances with a forward radiative transfer model. However, while the observed radiances can be perturbed by 3D cloud effects, the forward model tries to match those radiances with an independent pixel approximation that does not physically include 3D cloud effects. In particular the 3D cloud effect enhances, or brightens, the radiances as compared to no clouds being present. To compensate for this brightening the forward model decreases the retrieved surface pressure (reduction in dp_abp), increases the optical depth of cloud water (aod_water) and increases the surface albedo in the $WCO_2$ band. These relationships are shown empirically in Figure 7 As shown in Fig. 2 of Massie et al. (2021), the spectral signature of the 3D cloud effect (the optical depth structure of the radiative perturbation of the 3D effect) differs from the spectral signatures of perturbations in surface pressure, surface reflectivity, aerosol, and $X_{CO2}$. Fig. 2 illustrates that a decrease in surface pressure and $X_{CO2}$, and an increase in surface reflectance will increase the observed radiance. In order to provide for extra radiance enhancement in the cloud brightened observed radiance, a variety of state variable adjustments (and their unique spectral contributions) are utilized by the retrieval to bring forward model radiances in agreement with the observed radiances. The relationship of 3D cloud biases to surface pressure differences and surface albedo are likely due to a combination of physically-based 3D cloud radiative effects and operational retrieval algorithmic considerations. "*

**Specific comments**

Introduction: This is a too short introduction. It is purely focussed on OCO-2. What is missing is an overview of the OCO-2 algorithm and the relation with real 3D cloud effects. There is no mention of shadows. No mention of other literature in the subject is dealt with.

*Added information about the OCO-2 algorithm, the physics of the 3D cloud effect, and how it interacts with the retrieval:*

*L27 "Using an optimal estimation retrieval (Rodgers, 2000) called ACOS (O'dell et al., 2018), these measurements are converted to column-averaged atmospheric $CO_2$ dry-air mole fractions ($X_{CO2}$). ACOS employs a physics-based forward model that takes into consideration viewing and solar geometry and various atmospheric and surface parameters. Since OCO-2 generates on the order of 100,000 soundings per day, ACOS makes multiple approximations to speed up the retrieval algorithm. Most importantly, the retrieval makes the independent pixel approximation, where the radiance in a given sounding only depends on the properties (e.g. surface reflectance, aerosols, trace gas concentration) within the field of view of this sounding. **This approximation exploits that for most clear sky observations there is no significant horizontal exchange of photons. However, nearby clouds can scatter a significant number of photons into the field of view of OCO-2 which enhances the observed radiance. This horizontal exchange of photons due to clouds, or 3D cloud effect, is not accounted for in the ACOS retrieval. Nevertheless, the forward model attempts to match the enhanced radiances by modifying state vector variables (such as surface pressure, surface reflectance, aerosol, XCO2, and other variables). Theoretical calculations in Massie et al. (2021) indicate that the spectral signatures of changes in radiance due to changes in surface pressure, surface reflectance, aerosol, XCO2, and the radiative signature of the 3D enhancement, are different. The operational retrieval adjusts the state vector elements to enhance the 1D radiances to match the observed 3D radiance. This adjustment process, however, is imperfect (many state vector elements are perturbed) which leads to errors in the converged state vector and most importantly,** negative biases in retrieved $X_{CO2}$ (Massie et al., 2021; Massie et al., 2017; Merrelli et al., 2015; Emde et al., 2022; Kylling et al., 2022; Yu et al., 2021)."*

*On real cloud effects,*

*L39 "Merrelli et al. (2015) applied the Spherical Harmonics Discrete Ordinate Method (SHDOM) 3D radiative transfer code (Evans 1998) to perturb OCO-2 type spectra, and calculated OCO-2 retrievals without and with the 3D radiance perturbations. Retrieved XCO2 values were lower than clear sky retrievals by 0.3, 3, and 5-6 ppm for surfaces characterized by bare soil, vegetation, and snow-covered footprints. From an empirical perspective, Fig. 6 of Massie et al. (2021) demonstrates that ocean $X_{CO2}$ generally decreases in value when the distance between observations and clouds becomes less than 5 km."*

*On cloud shadows,*

*L45 "Nearby clouds can also cause radiance dimming due to cloud shadows. But only 25 % of observed radiances pass into the operational retrieval, since two cloud pre-processors (Taylor et al. 2016) exclude many observed radiances. Cloud brightening occurs on both sides of clouds since 40% of OCO-2 observations are within 4 km of clouds (Massie et al. 2021), and cloud brightening extends over a 5 to 10 km horizontal scale. A cloud shadow occurs only on one side of a cloud, with the shadow covering a limited angular portion of the side. Since the majority of OCO-2 observations are next to low-level clouds (think of an observation embedded in low-level Amazon cloud streets), the cloud shadows project only about a km or so from the low- level clouds. Using a year's data volume, Massie et al. (in prep.) discuss detailed calculations, based on an analysis of OCO-2 O2 A-band continuum radiances, that yield an estimate of cloud shadowing frequency to be on the order of 4%, compared to 96% for the observations influenced by cloud brightening."*

1. l. 35: B10 is used later on as a jargon word for the new data version of XCO2.

   *Replaced mentioning of 'B10', throughout most of the manuscript, by 'OCO-2' or omitted where possible*

2. l. 37: does this include overpasses over strong sources or plumes of CO2?

   *Yes, the 'small areas analysis' includes CO2 plumes that are erroneously attributed to biases in the retrieval. However, they are rare in the dataset and therefore should not impact the bias correction model significantly. To demonstrate that CO2 plumes are not being removed by our developed model we added an analysis of three power plant flyovers by OCO-2 to Section 5.3*

3. l. 50 ff (at many places): B10 lite files: please write out in normal words, since this is jargon.

   *Changed sentence to: "We make use of OCO-2 (B10) (https://disc.gsfc.nasa.gov/datasets/OCO2_L2_Lite_FP_10r/, last access: 05/2022) data from September 2014 to July 2019."*

   *Replaced mentioning of 'B10', throughout most of the manuscript, by 'OCO-2' or omitted where possible*

4.  l. 62-65: the three "3D cloud effect features" H3D, HC, and CSNoiseRatio should be explained: what does the acronym mean, and what is the definition. Also the definition of Cloud Distance should be given.

*L82 "Finally, we make use of four variables indicative of 3D cloud effects (Massie et al., 2021): H3D, HC, CSNoiseRatio, and Cloud Distance. H3D (Liang et al., 2009; Massie et al., 2017) describes the normalized standard deviation of the MODIS radiance field, and is calculated based on the Cronk (2022) off-line MODIS radiance data files. The radiance standard deviation is calculated in a circle of radius 10 km surrounding each OCO-2 data point. HC is calculated from differences in O2 A-band continuum radiances of an observation point and adjacent points in three rows (frames) of footprints. A frame has eight adjacent OCO-2 footprints, with each footprint on the order of 2 km in size. CSNoiseRatio is the ratio of the O2 A-band continuum radiance spatial standard deviation and noise level, calculated within a footprint (which has 20 "colorslice, CS" sub-pixel elements). These three variables are indicative of 3D cloud effects since radiance gradients are present when clouds are next to observation footprints (radiance enhancements become larger as cloud distance decreases). Cloud Distance (Massie et al., 2021) is the distance of the nearest cloud to each observation point, as determined from the Cronk (2022) off-line radiance data files, which contain 500 m MODIS radiances, geolocation and cloud mask data. Calculated 3D cloud features can be found for OCO-2 from September 2014 to July 2019 at https://doi.org/10.5281/zenodo.4008764."*

5.  l. 77-78: unclear sentence, please reformulate.

*Reformulated sentence to:*

*L103 "For each small area we define the median retrieved $X_{CO2}$ of this small area as the true $X_{CO2}$ and any differences to this median are treated as biases."*

6.  l. 95-96: how do you know that the long tail is indicative of 3D cloud effects?

*Previous work by Massie et al. (2021) showed that 3D cloud effects cause negative biases in XCO2 and dominate low quality OCO-2 data (with QF=1). Therefore, we describe the long tail of negative biases that in XCO2, most notably for QF=1 data, as indicative of 3D cloud effects.*

*L139 Added reference of the paper.*

7.  l. 118: form > from

*corrected*

8.  l. 122: ridge regression: please give a reference.

*Added citation for ridge regression: (Hoerl & Kennard, 1970a, 1970b)*

9.  l. 125, Equation 1: What is the meaning of the subscripts '2' in this formula?

*The subscript 2 in this context stands for the Euclidean norm. Added that to the manuscript:*

*L173 "... **X** are the standardized features, $\|\cdot\|_2$ is the Euclidean norm, and α controls the strength of the Tikhonov regularization "*

10. l. 138: more information can be found …: you refer to an internal technical report of many pages - please give here the relevant information of each key variable.

    *Added the information for which pages contain that information:*

    *L182 "This results in a set of 23 features for soundings over land, and 24 features for soundings over sea, that may be used to correct for 3D cloud biases in retrieved $X_{CO2}$ (more information about each variable can be found on pages 29 to 40 in (Jet Propulsion Laboratory, 2018))"*

11. l. 152: robust relationship to 3D cloud biases: how do you know? are these biases not due to other retrieval model errors than 3D?

    *Removed that statement and added a note that separating biases due to 3D cloud effects and other causes is challenging:*

    *L217 "Note that it is not possible to clearly separate biases due to 3D cloud effects and other mismatches between the forward model of the retrieval algorithm and the observed radiances. For example, differences in modelled and real aerosol optical properties (Chen et al., 2022) or uncertainties in absorption profiles of various trace gases (Payne et al., 2020) likely are important. Additionally, uncertainties in the instrument calibration can cause systematic biases as well. Thus, some of the features might also correct for non-3D cloud effects. However, we tried to mitigate the effect of non-3D cloud biases by only adding features to the feature selection process that show some dependence to nearest cloud distance (see Figure 2) or have a direct physical relationship to 3D cloud biases. Additionally, our bias correction is applied to data that has already been corrected with the operational OCO-2 bias correction (our processing utilizes bias corrected $X_{CO2}$). Thus, biases independent to 3D cloud effects should be minimized."*

12. l. 158-159: What does this finding mean? Please discuss if this is good or bad news.

    *Added the following discussion:*

    *L208 "This indicates that elements of the operational retrieval state vector (co2_grad_del, dp_abp, h2o_ratio, albedo_wco2, albedo_sco2) are more directly correlated with remaining biases in $X_{CO2}$ (due to 3D cloud and other effects) than features that directly measure 3D cloud effects which perturb the radiation field (H3D, HC, CSNoiseRatio). From an operational standpoint, using elements from the current retrieval state vector to correct 3D cloud biases simplifies the bias correction in future operational products. It also is more generally applicable to other missions that might not have available coincident cloud field measurements, that can be applied to derive nearest cloud distances, such as OCO-3 (Eldering et al., 2019). On the other hand, it reduces the interpretability of the developed model and does not allow to directly link 3D cloud biases to 3D cloud metrics. The OCO-2 and 3D cloud variables and their meaning are summarized in Table 2."*

13. l. 160-161: how do you know that albedo_wco2 has a direct connection to 3D cloud effects?

*3D cloud effects stem from the horizontal exchange of photons due to clouds. Surface albedo impacts this horizontal exchange since more photons can get reflected that make it eventually to the OCO-2 sensor. Added the following explanation to the manuscript to clarify the statement:*

*L180 "Those variables have a direct physical impact on 3D cloud effects; 3D cloud effects are amplified at large solar zenith angles and for brighter surfaces (Okata et al., 2017)."*

14. l. 224-226: Isn't this finding contradicting the title of the manuscript? It appears that the biases are not due to clouds.

*No, it only means that other variables are more directly correlated with biases in XCO2 that, at least in part, originate from 3D cloud effects. That is already stated in the manuscript:*

*L297 "As shown in Figure 3, the four cloud variables (H3D, HC, CSNoiseRatio, Nearest Cloud Distance) were removed during the recursive feature elimination step, indicating that other variables form the state vector are more directly correlated with $X_{CO2}$ biases."*

15. l. 239-243: this is unreadable text. Please give a table of features and their meaning.

*Added table of the variables used in the manuscript:*

**"Table 2: Summary of OCO-2 state vector variables and 3D cloud variables**

| Variables | Description |
|---|---|
| dp_abp | Retrieved surface pressure minus surface pressure from forecast model |
| h2o_ratio | Ratio of retrieved $H_2O$ column from $WCO_2$ band to that from $SCO_2$ band |
| co2_grad_del | Change between the retrieved $CO_2$ profile and the a priori profile |
| aod_water | Retrieved extinction optical depth of cloud water |
| albedo_sco2 | Retrieved surface albedo in $SCO_2$ band |
| albedo_wco2 | Retrieved surface albedo in $WCO_2$ band |
| | |
| H3D | Normalized standard deviation of the radiance field |
| HC | Differences in continuum radiances of an observation to adjacent observations |
| CSNoiseRatio | Ratio of the continuum radiance spatial standard deviation and noise level |
| Cloud Distance | Distance to the nearest cloud |

*"*

16. l. 304: xco2_strong_idp: this is jargon. Please use understandable words or acronyms in a sentence. The same holds for the other "input features".

*Replaced names of features with more general terms where appropriate and remind the reader in the Discussion section what the individual variables stand for.*

*L366 "**Differences between retrieved surface pressure and surface pressure from a forecast model (dp_abp)** show a positive correlation with $X_{CO2}$ biases. When the operationally retrieved surface pressure is underestimated, $X_{CO2}$ is underestimated as well. **The ratio of retrieved H2O column from WCO$_2$ band to that from SCO$_2$ band (h2o_ratio)** for soundings over land (Figure 7b) is independent to $X_{CO2}$ biases for ratios of less than one and has a strong negative correlation for ratios above one. A ratio of 1.05 corresponds on average to an $X_{CO2}$ bias of -1 ppm. **The difference between the retrieved CO$_2$ profile and the a priori profile (co2_grad_del)** shows mostly a positive correlation for negative values (surface CO$_2$ is underestimated compared to CO$_2$ higher up in the atmosphere) and a negative correlation for positive values. This indicates that 3D cloud effects challenge the accurate retrieval of the $X_{CO2}$ profile. The sensitivity of $X_{CO2}$ biases to changes in co2_grad_del is approximately twice as strong over sea than land (see Figure 7c and 7f). **The retrieved extinction optical depth of cloud water (aod_water)** shows a mostly negative linear correlation with a $X_{CO2}$ bias of -1 ppm for an extinction optical depth of 0.1. **Finally, the surface albedo in the weak and strong CO$_2$ band (albedo_wco2, albedo_sco2)** have mostly no dependence to $X_{CO2}$ biases for most of their range but show some positive and negative correlations with biases for brighter and darker surfaces, respectively."*

17. l. 306: what is the IDP preprocessor?

    *A preprocessor of the OCO-2 pipeline that performs a simplified XCO2 retrieval. We removed any mentioning of the preprocessor from the manuscript.*

18. l. 313-315: This is a very confusing statement. What is 3D effect physics and what is model mistake?

    *Simplified and clarified the statement:*

    *L379 "Note that our bias correction is applied in addition to the bias correction that has already been performed in the operational OCO-2 retrieval. While the operational OCO-2 bias correction does not explicitly account for 3D cloud biases it might implicitly mitigate such biases with its linear bias correction (since the operational bias correction variable dP is correlated to nearest cloud distance, see the red line in Fig. 2b)."*

19. l. 364: what are shallow angles? you do not mean small angles?

    *Changed "shallow" to "large"*

20. l. 375: please write this sentence in normal words.

    *Clarified the sentence. It now reads:*

    *L476 "We identified four variables from the state vector for OCO-2 retrievals over land (dp_abp, h2o_ratio, co2_grad_del, aod_water) and sea (dp_abp, co2_grad_del, albedo_sco2, albedo_wco2) that are used in a machine learning model that is able to remove 3D cloud biases in operational bias-corrected OCO-2 retrieved $X_{CO2}$."*

**Figures:**

Figure 1 (and following figures): please explain in the caption the acronyms used in the figure.

*Added explanation of acronyms to captions and simplified naming conventions in figures.*

Figure 3: Interesting figure. For both land and sunglint retrievals, the three features that are most important have no relation to clouds. What is the physics behind this? Please add a discussion of physics of the scene and retrieval model or process.

*Added explanation of how the individual variables relate to 3D cloud biases*

*L383 "To understand why some variables of the OCO-2 retrieved state vector are correlated with 3D cloud biases it is important to remember that the operational retrieval, based on optimal estimation, tries to match the observed radiances with a forward radiative transfer model. However, while the observed radiances can be perturbed by 3D cloud effects, the forward model tries to match those radiances with an independent pixel approximation that does not physically include 3D cloud effects. In particular the 3D cloud effect enhances, or brightens, the radiances as compared to no clouds being present. To compensate for this brightening the forward model decreases the retrieved surface pressure (reduction in dp_abp), increases the optical depth of cloud water (aod_water) and increases the surface albedo in the WCO$_2$ band. These relationships are shown empirically in Figure 7. As shown in Fig. 2 of Massie et al. (2021), the spectral signature of the 3D cloud effect (the optical depth structure of the radiative perturbation of the 3D effect) differs from the spectral signatures of perturbations in surface pressure, surface reflectivity, aerosol, and X$_{CO2}$. Fig. 2 illustrates that a decrease in surface pressure and X$_{CO2}$, and an increase in surface reflectance will increase the observed radiance. In order to provide for extra radiance enhancement in the cloud brightened observed radiance, a variety of state variable adjustments (and their unique spectral contributions) are utilized by the retrieval to bring forward model radiances in agreement with the observed radiances. The relationship of 3D cloud biases to surface pressure differences and surface albedo are likely due to a combination of physically-based 3D cloud radiative effects and operational retrieval algorithmic considerations."*

Figure 3: It is striking how many features do not add anything to R^2.

*It is very typical that a lot of features don't add any information to such a bias correction. Some variables are simply not correlated with biases in X$_{CO2}$ and other variables are correlated with each other. Thus, removing those variables will not change the resulting R$^2$.*

Figure 9: What is RF bias?

*RF bias is the bias in XCO2 as determined by our model. Simplified legend and replaced "RF bias" with "OCO-2 Bias"*

Figures 9 and 10: What is the correlation of the points in figures 9 and 10?

*The correlation of Nearest Cloud Distance and OCO-2 Bias is 0.3. Added this information to the manuscript:*

*L426 "When comparing the regional biases to a map of nearest cloud distance (see Figure 10) there is overlap between negative biases and areas dominated by clouds **(correlation coefficient between nearest cloud distance and OCO-2 bias is R=0.3)**."*

---

## Author Response (AR2)

We thank the anonymous reviewer for taking the time to review the revised manuscript a second time and providing helpful feedback. Your advice has been very helpful and allows to better link our truth proxy to 3D cloud effects.

The supplement contains the reviewer comments in black and our responses are shown in red *Italic* with citations from the manuscript in " ".

Reviewer 1:

Thank you for addressing the concerns raised in the review. The quality of the manuscript has been increased greatly.

My main remaining concern is the "truth metric", which uses the deviation from the local median in small area (maximum 100 km). This approach could underestimate the cloud-related bias, when the majority of OCO-2 pixels is in the vicinity of clouds (<10 km). To avoid such bias, it would be necessary to remove all pixels near clouds before computing the median. Since cloud distance is already available from the MODIS product, I think it should be easy to check if filtering for cloud distance affects the results and conclusions of this study.

*That is an excellent idea and links the truth proxy more directly to 3D cloud effects. Thanks for the suggestion. We recalculated the small areas truth metric as suggested, recomputed the various steps with our algorithm, and regenerated the figures and tables. While the overall conclusions are similar the previous approach slightly underestimated existing 3D cloud biases.*

*Discussion of the new truth proxy in 3.1 L96: 'As a pre-processing step we match the 3D cloud variables, OCO-2 soundings, and TCCON by time and location. Afterwards, we remove soundings where no 3D cloud variables are available. To develop the bias correction model, we use the small areas analysis, which is based on the assumption that CO2 is a well-mixed gas and assumed to be constant over spatial scales of less than ~100 km (though, there can be exceptions for strong CO2 emitters such as mega cities). To exploit this constraint on XCO2 we split OCO-2 soundings from the same orbit into small areas with a maximum size of 100 km. Each small area is generated by collecting soundings (sorted by observation time) until the distance between the first and last sounding exceeds the 100 km threshold. Afterwards, the collection process of the next small area is started. For each small area we identify soundings that are assumed to be free of 3D cloud biases (nearest cloud is at least 10 km away). From those soundings we define the median retrieved XCO2 as the true XCO2 of a given small area and any differences to this median are treated as biases. Small areas that contain less than 10 soundings free from 3D cloud biases are removed from the dataset. Since this process biases the remaining small areas towards longer cloud distances, we resample the remaining soundings so that the distribution of nearest cloud distances is similar to the original data set with about 40% of the sounding having a nearest cloud distance of less than 4 km.'*

Minor comments:

L45: The sentence is a bit unclear. Do you mean that 25% of all OCO-2 spectra are removed by the pre-processors that mainly look at clouds?

Removed that sentence to not confuse the reader.

L129f: Are the data "with QF=0 and QF=1" filtered here using the values from Table 1?

*That is correct. Changed the sentence to clarify this. It now reads in section 3.1 L110: 'Next, we remove outliers with large XCO2 errors from the data set by applying a series of thresholds to the variables from the state vector. The variables and their thresholds are given in Table 1. Note that these filters remove only a small fraction of soundings (4%).'*

L479: "operational bias-corrected OCO-2 retrieved XCO2": this is quite difficult to read.

*Agreed. Simplified the sentence to: '... allows to mitigate 3D cloud biases in OCO-2 retrieved XCO2.'*